# Comparative Proteomic Analysis Reveals the Regulatory Effects of H_2_S on Salt Tolerance of Mangrove Plant *Kandelia obovata*

**DOI:** 10.3390/ijms21010118

**Published:** 2019-12-23

**Authors:** Yi-Ling Liu, Zhi-Jun Shen, Martin Simon, Huan Li, Dong-Na Ma, Xue-Yi Zhu, Hai-Lei Zheng

**Affiliations:** Key Laboratory of the Ministry of Education for Coastal and Wetland Ecosystems, College of the Environment and Ecology, Xiamen University, Xiamen 361102, China; liuyiling@stu.xmu.edu.cn (Y.-L.L.); 33320130153898@stu.xmu.edu.cn (Z.-J.S.); martinsimon23@gmail.com (M.S.); huanlee723@stu.xmu.edu.cn (H.L.); madongna@stu.xmu.edu.cn (D.-N.M.); zhuxueyi90@xmu.edu.cn (X.-Y.Z.)

**Keywords:** *Kandelia obovata*, mangrove, hydrogen sulfide, salt tolerance, comparative proteome

## Abstract

As a dominant mangrove species, *Kandelia obovata* is distributed in an intertidal marsh with an active H_2_S release. Whether H_2_S participates in the salt tolerance of mangrove plants is still ambiguous, although increasing evidence has demonstrated that H_2_S functions in plant responses to multiple abiotic stresses. In this study, NaHS was used as an H_2_S donor to investigate the regulatory mechanism of H_2_S on the salt tolerance of *K. obovata* seedlings by using a combined physiological and proteomic analysis. The results showed that the reduction in photosynthesis (Pn) caused by 400 mM of NaCl was recovered by the addition of NaHS (200 μM). Furthermore, the application of H_2_S enhanced the quantum efficiency of photosystem II (PSII) and the membrane lipid stability, implying that H_2_S is beneficial to the survival of *K. obovata* seedlings under high salinity. We further identified 37 differentially expressed proteins by proteomic approaches under salinity and NaHS treatments. Among them, the proteins that are related to photosynthesis, primary metabolism, stress response and hormone biosynthesis were primarily enriched. The physiological and proteomic results highlighted that exogenous H_2_S up-regulated photosynthesis and energy metabolism to help *K. obovata* to cope with high salinity. Specifically, H_2_S increased photosynthetic electron transfer, chlorophyll biosynthesis and carbon fixation in *K. obovata* leaves under salt stress. Furthermore, the abundances of other proteins related to the metabolic pathway, such as antioxidation (ascorbic acid peroxidase (APX), copper/zinc superoxide dismutase (CSD2), and pancreatic and duodenal homeobox 1 (PDX1)), protein synthesis (heat-shock protein (HSP), chaperonin family protein (Cpn) 20), nitrogen metabolism (glutamine synthetase 1 and 2 (GS2), GS1:1), glycolysis (phosphoglycerate kinase (PGK) and triosephosphate isomerase (TPI)), and the ascorbate–glutathione (AsA–GSH) cycle were increased by H_2_S under high salinity. These findings provide new insights into the roles of H_2_S in the adaptations of the *K. obovata* mangrove plant to high salinity environments.

## 1. Introduction

Mangroves comprise a complex and unique ecosystem that distributes along tropical and subtropical coastal tidal zone [1]. Mangroves are facultative halophytes and potential stress adaptors due to their special morphological, anatomical, physiological, and biochemical features [1]. Different mangrove species have different salinity preferences and achieve optimal growth at different salinity levels. Jayatissa et al. reported that there is an optimal growth for *Sonneratia caseolaris* with the lowest salt tolerance at 3–5 ppt salinity, and there is an optimal growth for *Avicennia marina* with the highest salt tolerance at 25–27 ppt salinity [2]. As a dominant species of mangrove plants, *Kandelia obovata* can distribute in areas with salinities up to 27.58 ppt seawater level [3]. The salinity between 5 and 15 ppt is suitable for indoor-cultured *K. obovata* growth, while salinity up to 20 ppt inhibits the photosynthesis and growth of *K. obovata* [4].

Tidal inundation, which originates from coastal waters, influences sediment salinity and pore-water sulfide concentration [5]. Soil salinities and porewater sulfide concentrations are further enhanced with high inundation [6]. Hydrogen sulfide (H_2_S) is the dominant sulfide-containing gas that is emitted from intertidal sediment. A mangrove forest was found to serve as a perpetual source of H_2_S, and the annual mean emission of H_2_S found in this study was 768 ± 240 μg·S·m^−2^·d^−1^ [7]. Mangrove plants affect the sulfide concentration and H_2_S emission flux of sediments [8], and this research focuses on whether H_2_S influences the adaptation of mangrove plants to harsh environmental conditions such as high salinity.

As seen in huge body of previous studies, plants gain their tolerance to abiotic stress by actively synthesizing H_2_S. Hou et al. reported that drought stress leads to stomatal closure through an increase in the expression and activity of d/l-cysteine desulfhydrase (d/l-CD), a key enzyme of H_2_S biosynthesis in the leaves of *Vicia faba* [9]. The H_2_S pretreatment was found to significantly increase the regrowth ability of tobacco suspension-cultured cells after heat stress by alleviating a reduction in cell viability [10]. Salinity can detrimentally impact plant biomass production, which is associated with a salinity-induced a reduction in the amount of chlorophyll [11]. Previous studies have mostly examined the role of H_2_S in the mitigation of oxidative stress in salt-stressed plants [12,13,14]. The activities of superoxide dismutase (SOD), catalase (CAT), ascorbic acid peroxidase (APX), glutathione reductase (GR), glutathione peroxidase (GPX) and dehydroascorbate reductase (DHAR) were found to be increase in salt-stressed cucumber seedlings with the addition of NaHS, which is a widely used H_2_S donor, while, under the same experimental conditions, lipid peroxides and hydrogen peroxide (H_2_O_2_) levels were found to decrease [12]. The improved germination rate of salt-stressed cucumber seeds by H_2_S might have been due to the breakdown of starch by H_2_S-mediated α- and β-amylase induction in endosperm, which ultimately led to improved hypocotyl growth [13]. To salt-stressed strawberry, root pre-treatment with H_2_S has been found to induce an increase in stomatal conductance, photosynthesis, and leaf water content [14]. In salt-stressed rice, NaHS has been found to increase chlorophyll and protein content [15]. However, in a study by Koch et al., H_2_S negatively affected the anoxic production of energy and the energy-dependent N uptake in roots of the freshwater marsh species *Spartina alterniflora* [16].

In most true mangroves, the adaptations of seedlings to tolerate saline conditions have been widely studied [17]. However, the role of H_2_S in salinity tolerance improvement in mangroves is still mostly unknown. In this study, an appropriate concentration of NaHS, a donor of H_2_S, had a positive effect on *K. obovata* under high salinity considering both physiological and proteomic aspects. The *K. obovata* seedlings were exposed to 400 mM of NaCl with or without 200 μM of NaHS for seven days, and proteomics-based methodology (Two-dimensional (2-DE) accompanied by matrix-assisted laser desorption/ionization time-of-flight mass spectrometry (MALDI TOF MS)) was adopted to investigate the specific metabolic pathways and regulatory mechanisms of H_2_S on the salt tolerance of *K. obovata* seedlings.

## 2. Results

### 2.1. Effects of H_2_S and NaCl on Growth and Photosynthesis of K. Obovata Seedlings

As expected, exogenous NaHS treatments (varying from 50 to 350 μM) progressively alleviated the NaCl-induced decrease in inhibition on the leaf photosynthesis of *K. obovata* seedlings. On the other side, a high concentration of NaHS (500 μM) exhibited no beneficial effects (Figure 1). A powerful relationship was found among the observed parameters through two-tailed Pearson correlation and path analysis (Appendix A). Most of the physiological indices were positively correlated with the net photosynthetic rate (Pn) except intercellular carbon dioxide (Ci). The Pn had negative but significant associations (−0.655*) with Ci and displayed significant interactions with leaf dry weight (DW), chlorophyll content (Chl), stomatal conductance (Gs) and transpiration rate (Tr). Among different concentrations of NaHS, 200 μM was the most effective in relieving NaCl-induced adverse effects in *K. obovata* seedlings. The recovery rate of Pn at 200 μM was the highest compared with other concentrations (Appendix A). We used 200 µM of NaHS in the next experiments.

### 2.2. H_2_S Rather than other Derivatives from Nahs Alleviates NaCl-Induced Reduction in Chlorophyll Content

It should be noticed that the NaHS solution contained not only H_2_S/HS^−^ but also other sulfur-containing components. The research results showed that the above chemicals (Na_2_SO_4_, NaHSO_3_, NaHSO_4_, and CH_3_COONa), failed to rescue the NaCl-induced reduction of chlorophyll content (Appendix A). Meanwhile, compared with the experimental control group (CK), the endogenous H_2_S content kept a high level with the NaHS treatment but maintained a stable low level under other sulfur-containing compounds or sodium-containing compounds (Appendix A). HS^−^ and/or H_2_S has an essential role in relieving the reduction in chlorophyll content in *K. obovata* seedlings under salt treatment but not other sulfur-containing compounds or sodium-containing compounds (Na_2_SO_4_, NaHSO_3_, NaHSO_4_, and CH_3_COONa).

### 2.3. Effects of H_2_S and NaCl on the Characteristics of Chlorophyll Fluorescence

The variable fluorescent/maximal fluorescent (*F_v_/F_m_*) ratio reflects the potential quantum efficiency of photosystem II (PSII), widely used as a sensitive indicator of stress-induced damage to the photosynthetic apparatus [18]. After seven days of treatment with NaCl, *K. obovata* seedlings’ leaves were used to measure the chlorophyll fluorescence. The results showed that the NaCl treatment significantly decreased the ratio of *F_v_/F_m_* by 27.18% compared to the control. However, when exogenous H_2_S was applied to the NaCl-treated seedlings, the *F_v_/F_m_* was increased by 16.8% (Appendix A).

### 2.4. Effects of H_2_S and NaCl on Oxidative Stress and Activity of the Antioxidant System

Cytomembranes are damaged by reactive oxygen species (ROS) that were induced by salinity. The content of H_2_O_2_ was analyzed to evaluate the effect of H_2_S on the oxidative burst of NaCl-treated *K. obovata* seedlings. When exogenous H_2_S was applied to NaCl-treated leaves, the content of H_2_O_2_ was decreased, but it did not significantly differ from that of the NaCl treatment (Figure 2A). Leaf electrolyte leakage percentage (REL) and the thiobarbituric acid reactive substances (TBARS) accumulation were measured to estimate the protective role of H_2_S on the stability of the membrane. The TBARS levels were decreased by the exogenous H_2_S addition, but these levels did not significantly differ from that of the NaCl treatment (Figure 2B). Meanwhile, high salinity significantly increased the REL of *K. obovata* leaves, being 2.0-fold higher than that of the control. The exogenous H_2_S significantly alleviated the NaCl-induced increase in REL (Figure 2C).

Additionally, the content of glutathione (GSH) significantly increased by around 4-fold after the NaCl and H_2_S treatment compared to the NaCl treatment (Figure 2D). Plant cells have evolved the antioxidant enzymatic system, such as SOD and APX, to cope with stress-induced ROS generation. When exogenous H_2_S was applied to NaCl-treated seedlings, the activity of APX and SOD were increased but did not significantly differ from that of the NaCl treatment (Figure 2E,F).

### 2.5. Identification and Classification of DEPs

The 2-DE proteomic approach was employed to analyze the protein expression profile to investigate the mechanism of the H_2_S-mediated salt tolerance of *K. obovata*. Three biological replicates were performed with similar results. According to the image analysis by the PDQuest software, approximately 450 protein spots were reproducibly resolved on each gel. The protein spots were detected in the isoelectric point (pI) range from 4 to 7, and the range of molecular weight (MW) was 14.4–66.2 kDa. A total of 37 spots with significant changes (higher than 2.0-fold, *p* < 0.05) were identified as differentially expressed proteins (DEPs) and presented in representative image (Figure 3A). Close-up images from four protein spots were displayed in Figure 3B.

We provide the information for 37 DEPs in Table 1 and Appendix A. The difference between the two groups (NaCl vs. CK and NaCl and H_2_S vs. CK) was statistically significant. A comprehensive overview of the expression patterns of 37 identified proteins is shown in Figure 4A. They were divided into five functional categories, including carbohydrate and energy metabolism (29.73%), photosynthesis (18.92%), stress response proteins and cell structure (18.92%), amino acid and protein metabolism (16.22%), and hormone biosynthesis and transcription factor (10.81%) by their biological functions, each of which is in the UniProt database (Figure 4B). There were fifteen proteins that significantly changed under the NaCl treatment; specifically, the abundances of seven proteins were increased, and the abundances of eight proteins were decreased. Under the NaCl and H_2_S treatment, fourteen proteins were significantly changed, the abundances of nine proteins were increased, and five proteins were decreased. There were six proteins that were affected by both the NaCl and NaCl and H_2_S treatments (Figure 4A). Compared with the NaCl vs. CK group, there were more up-regulated DEPs in the NaCl and H_2_S vs. CK group. These up-regulated DEPs were mostly divided into carbohydrate metabolism, energy metabolism, and stress response proteins (Figure 4B). There were less down-regulated DEPs divided into carbohydrate metabolism, energy metabolism, and photosynthesis under the NaCl and H_2_S treatment (Figure 4B).

### 2.6. STRING Interaction Networks among DEPs

To predict protein–protein interaction (PPI) networks with the Search Tool of the Retrieval of Interaction Genes/Proteins (STRING) database, an alignment of the amino acid sequence to find the corresponding homologous proteins from *Arabidopsis thaliana* was carried out. The PPI network visualized by Cytoscape is shown in Appendix A. The PPI network of DEPs was found to be composed of 27 nodes and 129 edges. A total of two modules were identified in the PPI network, and these modules include nine and eight proteins (Figure 5, Appendix A). Module A has nine nodes and 32 interactions. The key node in module A is glutamine synthetase 2 (GS2). Module B has eight nodes and 22 interactions. The key node in module B is ribulose-1,5-bisphosphate carboxylase large subunit (RBCL).

An acyclic graph was constructed and directed by using the Biological Network Gene Ontology tool (BiNGO) to depict the visual interactions of functions based on the enrichment levels of the GO terms. The present study identified eight significantly enriched GO terms in biological processes, and especially in terms of response to abiotic stimulus, response to stress, and the generation of precursor metabolites and energy (Appendix A).

### 2.7. Comparisons of Expression Patterns between Protein and Transcript of Selected DEPs

To further verify the results of the proteome, we randomly selected six DEPs to further analyze their transcriptional expressions under different treatments by using quantitative real-time PCR. In particular, the consistency between the protein and mRNA levels of the following proteins was compared: transcription factor basic helix-loop-helix 145 (bHLH145, spot 13), superoxide dismutase 4 (SOD, spot 18), heat-shock protein (HSP, spot 19), oxygen-evolving enhancer protein 1 (OEE1, spot 14), abscisic acid stress ripening protein (Asr, spot 31) and glutamine synthetase 1;1 (GS1;1, spot 33). The detailed list of the corresponding primer pairs and the detailed PCR procedures used for the real-time PCR are summarized in Appendix A. As shown in Figure 6, four selected genes showed a similar pattern with their corresponding protein, except for bHLH 145 and GS1;1.

## 3. Discussion

### 3.1. H_2_S but Not Other NaHS Derivatives Contributes to Chlorophyll Content Recovery Decreased by NaCl Treatment

NaHS, which in solution dissociates to Na^+^ and HS^−^ (of which the latter associates with H^+^ to produce H_2_S), is widely used to examine the biological effects of H_2_S [19]. In contrast to the results from NaHS, treatment with other compounds failed to induce endogenous H_2_S accumulation and to alleviate the reduction in chlorophyll content caused by NaCl exposure (Appendix A). Accordingly, this study provides evidence that NaHS contributes to chlorophyll content recovery by HS^−^, rather than Na^+^. NaHS-associated responses are H_2_S-specific in *K. obovata*.

### 3.2. Exogenous H_2_S Alleviates Growth and Photosynthesis Inhibition Induced by NaCl in K. obovata Seedlings

Parameters such as relatively low fresh weight, relatively low dry weight, and reduced photosynthesis are potential indicators of salinity stress in plants [11]. In this study, significant variations in the studied physiological traits of *K. obovata* under different concentrations of NaHS with 400 mM of NaCl were recorded. When the concentration of H_2_S reached 500 μM, the leaf dry weight and photosynthesis decreased (Figure 1). These findings are in agreement with those of Chen et al. [20], who reported a similar decline in the biomass and photosynthesis of *Spinacia oleracea* after treatment with a high concentration of NaHS. The change in Ci was analyzed as an indicator of the dominant factor of stomatal or non-stomatal limitations to photosynthesis. Ci showed a reverse trend to Gs, which suggests that non-stomatal closure is the dominant limitation to photosynthesis [21]. In the present research, the decrease in Ci showed a reverse trend to Gs under various concentrations of NaHS, which indicated that non-stomatal limitations dominated.

The previous study indicated that H_2_S could indirectly enhance the photosynthesis reaching its maximal values, as well as the maximal photochemical efficiency of photosystem II (*F_v/_F_m_*) [20]. In the present research, the NaCl and H_2_S treatment also increased *F_v/_F_m_* (Appendix A). Additionally, the proteomics analyses revealed six proteins involved in the light reactions of photosynthesis, including electron transfer, light-harvesting, and the light-induced oxidation of water. The level of ferredoxin-NADP reductase (FNR, spot 2), which catalyzes the removal of the electron from photosystem I to NADP^+^, and otherwise nicotinamide adenine dinucleotide phosphate (NADPH) is oxidized to produce reducing power via ferredoxin (Fd) [22], which is recovered under the NaCl and H_2_S treatment. The O_2_ evolving complex 33kD family protein (OEC spot 12) has been termed as a chloroplast manganese-stabilizing protein due to its functions in assembly, stabilization and protection of the manganese cluster required for oxygen evolution in higher plants [23]. Though the accumulation of OEC was down-regulated in two groups, the accumulation of OEC was up-regulated relative to the NaCl treatment, which implied that H_2_S maintained the photosynthetic electron transport activity during *K. obovata* exposure to high salt stress. Additionally, NaCl-induced the down-regulation of photosynthetic electron transfer C (PETC, spot 10), which is an important subunit of the Cyt*b*_6_*f* complex that participates in linear electron transport [24], was alleviated by the addition of H_2_S. Nevertheless, plastocyanin (PC, spot 1) was up-regulated under NaCl, which is consistent with the study on *K. candel* under high salinity did by Wang et al. [25]. On another hand, the abundances of PC remained unchanged under the combined NaCl and NaHS treatment. The results of proteomics analyses are in agreement with physiological measurements that showed that the treatment with NaHS alleviated the NaCl-caused inhibition of photosynthesis.

Photosynthetic carbon assimilation is driven by ribulose-1,5-bisphosphate (RuBP) carboxylase (Rubisco). Ribulose-1,5-bisphosphate carboxylase large subunit (RBCL, spot 8) was down-regulated by the NaCl treatment and unchanged under the NaCl and NaHS treatment. It is noteworthy that ribulose activase small isoform precursor (RBCS, spot 36) showed a reverse trend that was unchanged under the NaCl treatment and down-regulated by the NaCl and NaHS treatment. The capacity of photosynthesis is associated with the level of the RBCL/RBCS ratio and the higher photosynthetic rate exhibited the higher RBCL/RBCS ratio in rice leaves [26]. The up-regulation of the RBCL/RBCS ratio by the NaHS addition under salt treatment in our study indicated that the alleviation of Rubisco inhibition inevitably leads to the improvement of photosynthesis.

Phosphoglycolate phosphatase (PGP, spot 25) is involved in photorespiration, which is required to give carbon from 2-phosphoglycolate back into metabolism [27]. The allocation of photosynthetic electron transport to photorespiration has been found to be enhanced with the decrease of the chlorophyll content [27]. In this study, the NaCl treatment strongly up-regulated PGP, which indicated that the allocation of photosynthetic electron transport to the Calvin cycle was reduced [28]. The accumulation of PGP was not significantly different from the control treatment by the NaHS addition.

The photosynthetic electron transport chain converts light into chemical energy, supplying ATP and NADPH to drive the carbon dioxide reduction and fixation processes [29]. ATP synthase F1 complex (CF1) epsilon subunit (atpC, spot 7) is involved in the energy supply needed for the carbon dioxide reduction and fixation processes. Though the accumulation of atpC was down-regulated in our study, the accumulation of atpC was modified relative to the NaCl treatment, thus implying the NaHS addition might have had a positive effect to supply the energy.

### 3.3. H_2_S Rescues the Primary Metabolism Altered by NaCl

In this study, triosephosphate isomerase (TPI, spots 16), and phosphoglycerate kinase (PGK, spot 35) were found to be down-regulated or unchanged under high salinity, respectively. PGK plays a crucial role in catalyzing the ATP-dependent phosphorylation of phosphoglycerate in the Calvin cycle [30]. The two glycolytic enzymes were all positivity modulated by the NaHS addition, suggesting much more energy was produced by glycolysis.

All levels of plant function are affected by nitrogen metabolism [31]. In higher plants, glutamine synthetase (GS2 and GS1:1, spots 32 and 33, respectively) is a crucial enzyme of primary ammonium assimilation and nitrogen metabolism [32]. Glutamine synthetase participates in the synthesis of GSH through the glutamate biosynthesis pathway; thus, the overexpression of glutamine synthetase leads to more GSH formation [33,34]. It is interesting to note that the accumulation of glutamine synthetase was positivity modulated by the NaHS addition in our study.

HSP (spot 19) and 20 kDa chaperonin family protein (Cpn 20, spot 26) were involved in protein synthesis, a process that was positively modulated by the NaHS addition. As molecular chaperones, HSP and chaperonin family protein participate in protein transport, protein folding, and protein assembly processes [35]. During abiotic stress, chaperonins help plants combat increasing amount of incorrectly folded proteins [35].

Previous publications have reported that the TCA cycle provides essential precursors for amino acid biosynthesis and energy metabolism. ATP synthase subunit d (ATP5PD, spot 9) and ATP synthase beta subunit (ATP5F1B, spot 29) are located in the mitochondria. The ATP synthase is an important enzyme that catalyzes energy production by synthesizing ATP from ADP [36], providing energy to the TCA cycle. In mammals, increased oxidative stress represents an important factor for isolated disorders of ATP synthase [37], and there is a connection between ATP synthase and oxidative stress. Moreover, it has recently been predicted that a protein network in which ATP synthase is involved can contribute to protect plants against photo-oxidative damage [38]. In our study, the NaCl and H_2_S treatment significantly up-regulated the accumulation of ATP5PD and ATP5F1B. In addition, nucleoside diphosphate kinase 1 (NDPK1, spot 30), a housekeeping enzyme that maintains levels of CTP, GTP, and UTP in cells, was identified to be down-regulated under high salinity and significantly up-regulated by the NaHS addition. A recent study showed that NDPK grants transgenic plants tolerance to multiple stresses, including salt and extreme temperatures [39].

Remarkably, H_2_S rescued the primary metabolism altered by NaCl in our study. In a previous study, Li et al. also reported that increment in energy metabolism by H_2_S might be responsible for abiotic stress alleviation [40].

### 3.4. H_2_S Relieves Oxidative Stress Induced by NaCl

The salt-induced disruption of normal plant metabolism results in the accumulation of harmful ROS [41]. Excess ROS not only cause growth inhibition but also programmed cell death (PCD). The accumulation of H_2_O_2_ and consequent increase in malondialdehyde and REL are indicators of salt-induced oxidative damage to membranes [42]. In the present study, the NaHS pretreatment reduced the accumulation of REL under salt stress, which is in agreement with a previous study by Shi et al. [43]. In addition, Actin7 isoform 1 (Actin7, spot 23)—which is thought to aid in the stability of organelle membranes in plant cells and to induce PCD with the stabilization of actin depolymerization [44,45]—was significantly up-regulated under the NaCl and H_2_S treatment in our study. Meanwhile, plants trigger various enzymatic and non-enzymatic antioxidants to detoxify ROS and prevent cellular damage [46]. In this study, compared with the NaCl treatment, the content of GSH was found to be induced by application of H_2_S (Figure 2). Ascorbate (AsA) and GSH play important roles in the AsA–GSH cycle [47]. As an electron donor, APX reduces H_2_O_2_ into water by using AsA, and monodehydroascorbate can react with GSH to produce AsA and GSSG catalyzed by DHAR [47]. In our study, the increase in GSH content and the activity of APX indicate that H_2_S promoted the AsA–GSH cycle that was responsible for the removal of H_2_O_2_ under high salinity. In agreement, APX (spot 37) protein expression was found to be up-regulated under the NaCl and H_2_S treatment. Moreover, copper/zinc superoxide dismutase CSD2A-1 (CSD2, spot 17) was found to be up-regulated under the NaCl and H_2_S treatment, but SOD (spot 18) showed the opposite trend. The SODs were divided into four types according to the different metal cofactors in the catalytic site, and CSD2 is mainly found in higher plants [48]. We propose that CSD2 plays a critical role in a direct and rapid mechanism of ROS detoxification under high salinity. The SODs (spot 18) might be iron SOD, which was found in the cytoplasmic of plant or the bacterial cells and played no major role in *K. obovata* leaves [48]. In short, based on the above results, we can infer that H_2_S increased non-enzymatic antioxidants to detoxify ROS and to prevent cellular damage under high salinity.

Moreover, some abiotic stress-related proteins, such as cysteine proteinase (CP, spot 4), pyridoxine biosynthesis PDX1-like protein 3 (PDX1, spot11), and alcohol dehydrogenase (ADH, spots 20) were also found to respond to high salinity and the NaCl and H_2_S treatment. An increase in the CP activity occurs just before the first senescence symptoms became visible in most plants [49]. CP is involved in PCD, and the balance between the levels of CP and phytocystatins is kept in a certain state through the antagonistic activities of abscisic acid (ABA) and gibberellins [50,51]. In our study, the accumulation of CP was significantly down-regulated by the NaHS addition. PDX1 is regarded as pyridoxal phosphate-binding protein [52]. Pyridoxal phosphate-binding proteins protect plants from high ion concentrations by inducing cysteine biosynthesis or the synthesis of cystatin to inhibit PCD [53]. We observed that PDX1 was significantly up-regulated to inhibit PCD under the NaCl and H_2_S treatment. ADH catalyzes the regeneration of NAD^+^ from the reduction of acetaldehyde to ethanol [54]. H_2_S has been shown to inhibit ADH activity in the roots of the freshwater marsh species *Spartina alterniflora* [16], but the accumulation of ADH was down-regulated under the NaCl treatment in our study and did not change significantly with the NaHS addition. We observed that H_2_S helped alleviate the reduction of ADH under the NaCl treatment.

### 3.5. H_2_S Regulates Hormone Biosynthesis and Transcription Factor

Abscisic acid stress ripening protein (Asr, spot 31) is a downstream protein that participates in ABA signaling pathways [55]. The bHLH transcription factors serve as a negative feedback regulatory loop in ABA signaling in *Arabidopsis thaliana* [56]. In the present study, Asr and bHLH145 were up-regulated under the salt treatment and did not change significantly under the NaCl and H_2_S treatment, which indicated the signal transduction of ABA was inhibited by the addition of NaHS. Furthermore, in the promoter regions of the genes encoding CP exist ABA-responsive elements, and the transcription of CP is repressed by ABA through DOF (DNA binding One Zinc Finger) [51]. Additionally, NaHS up-regulated the accumulation of basic region-leucine zipper (bZIP, spot 27), which has been found to positively alter the adaptation of plants to adverse circumstances [57,58].

### 3.6. Protein–Protein Interaction Networks Analysis

The DEPs in the PPI network exhibited a higher enrichment, indicating a higher degree of modularization; therefore, the DEPs were divided into two modules to investigate their interactions by using the Molecular Complex Detection (MCODE) (Figure 5). In module A, chloroplastic drought-induced stress protein (CDSP), NAD(P)-binding Rossmann-fold superfamily protein isoform 1 (FLDH, spot 24), FNR, GS2, OEE, PC, PETC, and RBCS were enriched in response to stimulus. The key node in this module was GS2. In module B, six nodes (APX, Cpn20, TPI, PGK, 2-Cys peroxiredoxin (Prx), and SOD) were involved in response to abiotic stimulus, and another two nodes (RBCL, ATP5F1B) were related to the Calvin cycle. The two modules implied that response to abiotic stimulus influenced other biological processes.

## 4. Materials and Methods

### 4.1. Plant Growth and Treatment

Mature propagules of *K. obovata* were collected from the mangrove forest in the National Nature Reserve for Mangroves in Zhangjiang River Estuary (23°55′ N, 117°26′ E), Yunxiao County, Fujian Province, PR China, where salinity levels range from 8% to 20%. Healthy propagules of similar size (20 cm in length) were selected and pre-cultivated in a plastic pot (dimension of 12 cm in diameter and 11 cm in depth) with clean sands. The pots were placed in a growth chamber with a temperature of 25/30 °C (night/day), a relative humidity of 70%, and a photoperiod of 8 h dark/16 h light with around 1250 μmol m^−2^ s^−1^. A 1/4 strength Hoagland’s nutrient solution was used to cultivate the hypocotyls. The solutions were replaced every week.

The treatments of NaHS and NaCl were set up when the 3rd pair of leaves appeared for physiological assays. The healthy and uniform seedlings were randomly cultivated in three groups, and each group had three replicates.

In the first group, plants were supplied with NaHS and NaCl for 7 days. Different concentrations of NaHS at 0, 50, 100, 200, 350 and 500 µM (The average of H_2_S was 243.1 ± 234.9 μM in the pore water samples of mangrove forest sediment) [59] were prepared in the 1/4 strength Hoagland’s nutrient solution in the presence or absence of 400 mM of NaCl. After 7 days of treatment, photosynthesis, chlorophyll content, and dry weight were measured to determine the optimal concentration of NaHS for the salinity-induced growth inhibition.

The second group was supplied with a 1/4 strength Hoagland’s nutrient solution containing 400 mM of NaCl in the presence of NaHS (200 μM), Na_2_SO4 (200 μM), NaHSO_3_ (200 μM), NaHSO_4_ (200 μM) and CH_3_COONa (200 μM). After 7 days of treatments, the contents of chlorophyll and endogenous H_2_S were measured to distinguish the actual role of H_2_S/HS^−^ from the other compounds.

According to the results from above two experiments, the third group was supplied with a 1/4 strength Hoagland’s nutrient solution containing both 400 mM of NaCl (NaCl) and 400 mM of NaCl and 200 μM of NaHS (NaCl and H_2_S). The same volume of a 1/4 strength Hoagland’s nutrient solution was used as the control (CK). After chlorophyll fluorescence measurements, the second pair of leaves from the apex of the growing shoots was harvested, rapidly frozen in liquid nitrogen, and stored at −80 °C for a proteomic study and further biochemical analysis.

### 4.2. Measurements of Leaf Photosynthetic Pigment, Photosynthetic Rate and Chlorophyll Fluorescence Quenching

The contents of chlorophyll were measured according to the method of Lichtenthaler with little modifications [60]. Eighty percent (v/v) acetone was used to extract chlorophyll, and the content was estimated according to the absorbances at 470, 646 and 663 nm.

The net photosynthetic rate (Pn) of fully expanded leaves, as well as transpiration (Tr), stomatal conductance (Gs), and internal CO_2_ concentration (Ci), were determined by using a portable photosynthesis system (Li-6400, Li-Cor, Lincoln, NE, USA). Those measurements were carried out in the morning from 9:00 to 11:30. The recovery coefficient of Pn was calculated according to the following equation: Pn^NaCl^/Pn^CK^. Here, Pn^NaCl^ and Pn^CK^ stand for the leaf photosynthetic rate measured in the NaCl and CK treatments.

We applied a fluorometer for measuring chlorophyll fluorescence (Li-6400, Li-Cor, Lincoln, NE, USA). *Fv/Fm* was calculated by using (*F_m_* − *F*_0_)/*F_m_*, where minimum (dark) fluorescence (*F*_0_) was obtained by applying to measure light pulses at low frequency (0.03 μmol m^−2^ s^−1^ for 1 s). The maximum fluorescence (*F_m_*) was determined by applying a saturating light pulse (6000 μmol m^−2^ s^−1^ for 0.8 s) to a dark-adapted sample [61].

### 4.3. Determination of Dry Weight and the Content of Endogenous H_2_S

Endogenous H_2_S content was determined according to the procedure of Zhang et al. [62], and the method of Chen et al. was employed to determine the dry weight [63].

### 4.4. Measurements of Oxidative Stress and Antioxidant System Activity

The H_2_O_2_ content was measured according to the method of Hung and Kao [64]. Lipid peroxidation was determined by measuring thiobarbituric acid reactive substances (TBARS), according to the method of Yan et al. [65].

For the relative electrolyte leakage (REL) measurement, fresh leaves (0.2 g) were cut, added to deionized water (20 mL), and degassed for 10 min at room temperature. Using an electrical conductivity meter (DDS-11A), the initial reading of conductivity (E1) was recorded. Then, the solution containing plant materials was incubated at 100 °C for 15 min. The final reading of conductivity (E2) of the solution was recorded after cooling. Additionally, the reading of conductivity for deionized water (E0) was measured. Finally, the REL was estimated based on the following equation: REL (%) = (E1 − E0)/(E2 − E0) × 100%.

Superoxide dismutase (SOD) activity was determined according to the method described by Beauchamp and Fridovich [66]. Ascorbate peroxidase (APX) activity was estimated according to the method of Chen and Asada [67].

The total content of glutathione (GSH) was measured by a GSH kit (JBI, Nanjing, China). Leaves (0.3 g) were ground with 0.3 mL of H_3_PO_3_ (25%) and 0.9 mL of an NaH_2_PO_4_-EDTA buffer (0.1 M, pH 8.0). The homogenates were separated at 10,000× *g* at 4 °C for 20 min. The supernatant was used for GSH content measurement [68]. The absorbance at 420 nm was measured according to the manufacturer’s instructions.

### 4.5. Protein Extraction from K. Obovata Leaves and 2-DE PAGE and Image/Data Analysis

Proteins were extracted by a phenol extraction procedure and followed by methanolic ammonium acetate precipitation, according to the method described by Delaplace et al. [69]. We rehydrated 17 cm immobilized pH 4–7 gradient strips (BioRad, CA, USA) with 340 μL of a rehydration buffer (containing a 1 mg protein sample) in the tray overnight. An Ettan immobilized pH gradient (IPG)phor3 system (GE Healthcare Amersham Bioscience, Little Chalfont, UK) was used for isoelectric focusing (IEF), following the conditions described by Shen et al. [70]. In 12.5% acrylamide gels, we observed gel electrophoresis. Each experiment was repeated three times. Each treatment was performed for three biological replications for quantitative analysis.

We stained SDS-PAGE gels with Coomassie Brilliant Blue R-250, and these gels were scanned by Uniscan M3600 (China) at 600 dpi. Gels were analyzed by the PDQuest software (Version 8.0, Bio-Rad) [71]. Afterward, DEPs were obtained by pairwise comparison with a fold change ≥2.0 and a Student’s *t*-test (*p* < 0.05).

### 4.6. In-Gel Digestion, Identification and Classification of Differentially Expressed Proteins

We excised the DEPs with a more than 2.0-fold change from 2-DE gels [70]. Gel slices were incubated with 25 mM of NH_4_HCO_3_ and acetonitrile for four times to remove the color of coomassie brilliant blue-R250 (CBB-R250). Trypsin/Lys-C Mix (Promega, Madison, WI, USA) was used to digest the proteins. Trifluoroacetic acid (0.5–1%) was added to terminate digestion. The supernatant was collected at 12,000× *g* for 10 min. A MALDI-TOF-TOF mass spectrometer was used to analyze the supernatant for mass spectrometry identification (Applied Biosystems, Massachusetts, USA), according to the method of Hu et al. [71]. The search parameters were set as follows: database NCBInr (release date: 2018.12.01); taxonomy Viridiplantae (green plants); peptide mass ranged from 10 to 130 kDa; the coverage of protein sequence had to reach a minimum of 10%; proteins had scores higher than 60 (*p* < 0.05); results with a confidence interval % (C.I.%) value higher than 95% were considered to be an identification.

The functions of the identified proteins were determined in http://www.uniprot.org/uniprot, and then the proteins were divided into five groups according to their biological functions in the plant. The subcellular localization of identified proteins was obtained by the protein subcellular localization prediction tool (WoLF PSORT, http://wolfpsort.hgc.jp/), as well as from previously published papers when possible.

The STRING database was used to predict the protein–protein interactions (PPIs) [72], and Cytoscape was used to visualize significant protein–protein associations in the PPI network. A combined score of >0.4 was selected as the cut-off value. Then, module analysis was carried out by the Molecular Complex Detection (MCODE) plugin to highlight the biological significance of gene modules that respond to the NaCl and/or NaHS treatment in *K. obovata* leaves [73]. Subsequently, the BiNGO tool was used to visualize the level of enriched GO terms.

### 4.7. Quantitative Real-Time PCR Analysis

For the total RNA extraction, the frozen *K. obovata* leaves (0.1g) were ground in liquid nitrogen and extracted by using a Total RNA Kit (TaKaRa, Dalian, China). We observed the RNA quality and integrity with an ultraviolet spectrophotometer (Cary 50, Varian, USA) and agarose gel electrophoresis. The RNA was used to synthesized cDNAs with moloney murine leukemia virus (M-MLV) reverse transcriptase First-Strand cDNA synthesis kit (TaKaRa, Dalian, China), and the cDNA mixture was used as a template for subsequent PCRs. The primers used for real-time PCR are shown in Appendix A. A 10 μL real-time PCR mixture contained 2 μL of primers, 2 μL of cDNA, and 6 μL of SYBR Green (Sangon, Shanghai, China). Five independent biological replicates were used to perform gene expression. The relative gene expression was calculated by the 2^−ΔΔCT^ method, and actin was used as an an internal control [74]. The Bio-Rad iQ5 Multicolor Real-Time PCR Detection System (Bio-Rad, Hercules, CA) was used to run qRT-PCR.

### 4.8. Statistical Analysis

A two-tailed Student’s *t*-test (*p* < 0.05) and a Duncan’s multiple range test were used for statistical analysis. Pearson’s correlation among the amendment’s characteristics was evaluated with SPSS Statistics for Windows (Version 22.0, IBM Corp, Armonk, NY). The direct and indirect relationships among leaf drought weight, chlorophyll content, net photosynthetic rate, intercellular carbon dioxide, stomatal conductance and transpiration rate were evaluated through path analysis by using edgeR package. The heatmap of the DEPs was drawn by using the R software “heatmap package.”

## 5. Conclusions

Physiological and proteomic evidence in this study support beneficial role of H_2_S on *K. obovata* exposed to high salinity: (1) H_2_S increased carbon fixations and electron transfer, maintaining photosynthesis under high salinity; (2) H_2_S stimulated glycolysis to generate precursor metabolites, energy, and protein synthesis; (3) H_2_S increased non-enzymatic antioxidants to detoxify ROS and prevent cellular damage under high salinity; and (4) H_2_S affected the ABA signaling pathway, which led to a sustained plant adaptation to high salinity. As shown in Figure 7, we propose a working model to illustrate the detailed mechanism by which H_2_S alleviates NaCl-induced inhibition. In conclusion, these findings have important implications for understanding the functional role of H_2_S in the salt tolerance of *K. obovata*.

## Figures and Tables

**Figure 1 ijms-21-00118-f001:**
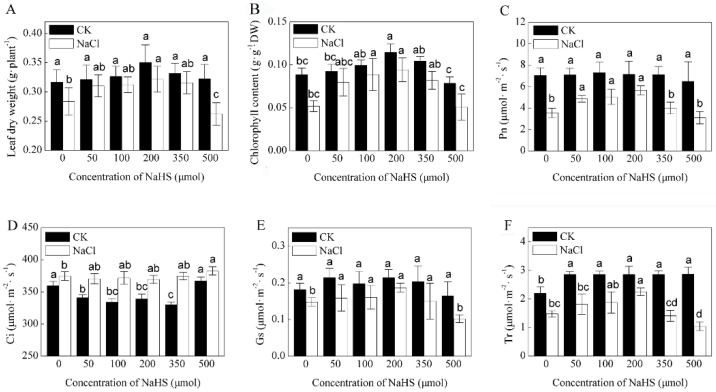
Effects of various concentrations of NaHS on the leaf photosynthetic characteristics of *Kandelia obovata* seedlings treated by high salinity. (**A**) leaf dry weight, (**B**) total chlorophyll contents, (**C**) net photosynthetic rate (Pn), (**D**) intercellular CO_2_ concentration (Ci), (**E**) stomatal conductance (Gs) and (**F**) transpiration rate (Tr). Seedlings were treated with different concentrations of NaHS (0, 50, 100, 200, 350 and 500 μM) together with 400 mM of NaCl for seven days. The sample without 400 mM of NaCl was regarded as the experimental control group (CK), and the sample with 400 mM of NaCl was regarded as the NaCl group. Values are means ± SE of three independent experiments with at least three replicates for each. Bars with different letters are significantly different at *p* < 0.05 according to Duncan’s multiple range test.

**Figure 2 ijms-21-00118-f002:**
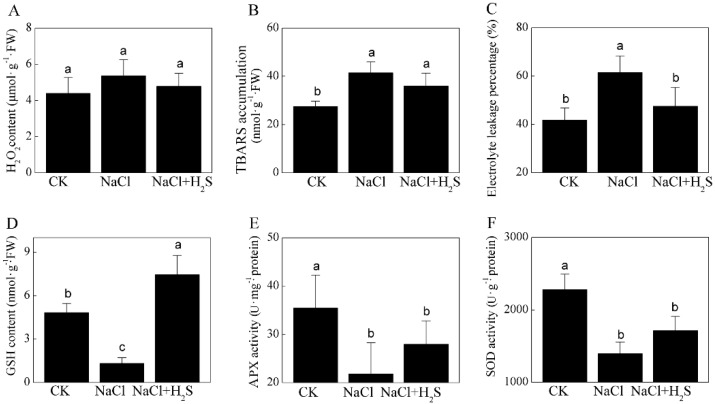
Effects of NaHS on oxidative stress and antioxidant system activity of the leaves of *K. obovata* seedlings treated by high salinity. (**A**) H_2_O_2_ content, (**B**) thiobarbituric acid reactive substances (TBARS) accumulation, (**C**) electrolyte leakage percentage (REL), (**D**) glutathione (GSH) content, (**E**) ascorbate acid peroxidase (APX) activity and (**F**) superoxide dismutase (SOD) activity. CK stands for the control treated only by a 1/4 strength Hoagland’s nutrient solution. NaCl stands for the 400 mM of the NaCl treatment. NaCl and H_2_S stands for the treatment with 400 mM of NaCl and 200 μM of NaHS. Error bars are SE (*n* = 3). The columns labeled with different letters are significantly different at *p* < 0.05 according to Duncan’s multiple range test.

**Figure 3 ijms-21-00118-f003:**
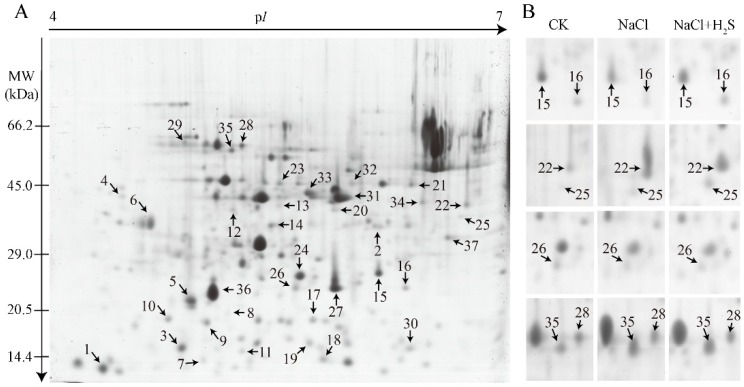
Two-dimensional (2-DE) analysis of proteins extracted from the leaves of *K. obovata* seedlings. The numbers assigned to the protein spots correspond to those listed in Table 1. (**A**) Representative coomassie brilliant blue-R250 (CBB-R250) stained 2D gel of total proteins extracted from *K. obovata* leaves under the control condition. Proteins (1.5 mg) were loaded onto the pH 4–7 gradient immobilized pH gradient (IPG) strip for the first-dimensional isoelectric focusing (IEF) and then separated in the second dimension on a 15% SDS-PAGE gel. The isoelectric point (pI) and molecular weight (MW) in kilodaltons are indicated on the top and left of the gel, respectively. Arrows indicate 37 spots with at least 2.0-fold changes (*p* < 0.05) that were analyzed by matrix-assisted laser desorption/ionization time-of-flight mass spectrometry (MALDI TOF MS). (**B**) The enlarged windows of the representative protein spots with different expressions under both the NaCl treatment and the NaCl and H_2_S treatment. NaCl stands for the 400 mM of NaCl treatment, NaCl and H_2_S stands for the treatment with 400 mM of NaCl and 200 μM NaHS, and CK stands for the control treated only by a 1/4 strength Hoagland solution.

**Figure 4 ijms-21-00118-f004:**
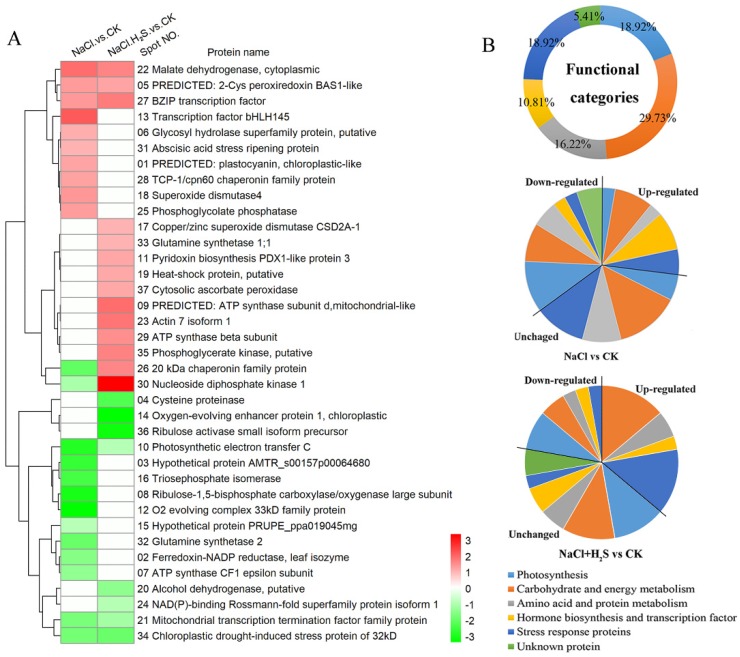
(**A**) Functional classification and hierarchical clustering analysis. (**B**) Functional categories for the 37 differentially expressed proteins in *K. obovata* seedling leaves under the NaCl treatment and the NaCl and H_2_S treatment. The rows represent the individual proteins. The protein cluster is on the left side, and the treatment cluster is on the top. The up- or down-regulated proteins are indicated in red or green, and white represents no change. The intensity of the color increases with increasing expression differences, as shown in the bar at the bottom of the figure.

**Figure 5 ijms-21-00118-f005:**
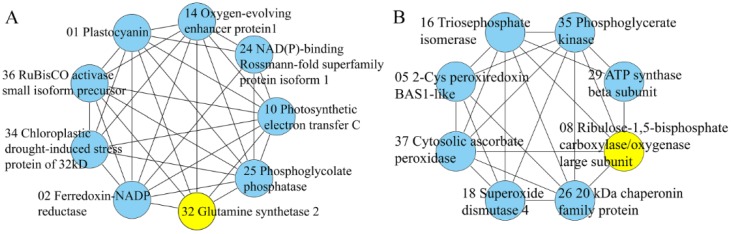
The two modules obtained from the protein–protein interaction network of differentially expressed proteins (DEPs). (**A**) Proteins in cluster 1 and (**B**) proteins in cluster 2.

**Figure 6 ijms-21-00118-f006:**
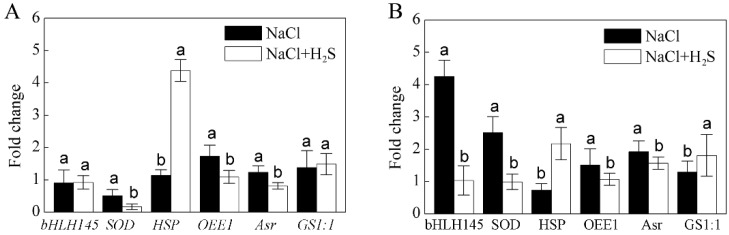
Comparison of expression changes at (**A**) mRNA and (**B**) protein levels for the selected six DEPs. They are basic helix-loop-helix 145 (bHLH 145), superoxide dismutase (SOD), heat-shock protein (HSP), oxygen-evolving enhancer protein 1 (OEE1), abscisic acid stress ripening protein (Asr), and glutamine synthetase 1;1 (GS1;1). The columns labeled with different letters are significantly different at *p* < 0.05 according to a Student’s *t*-test.

**Figure 7 ijms-21-00118-f007:**
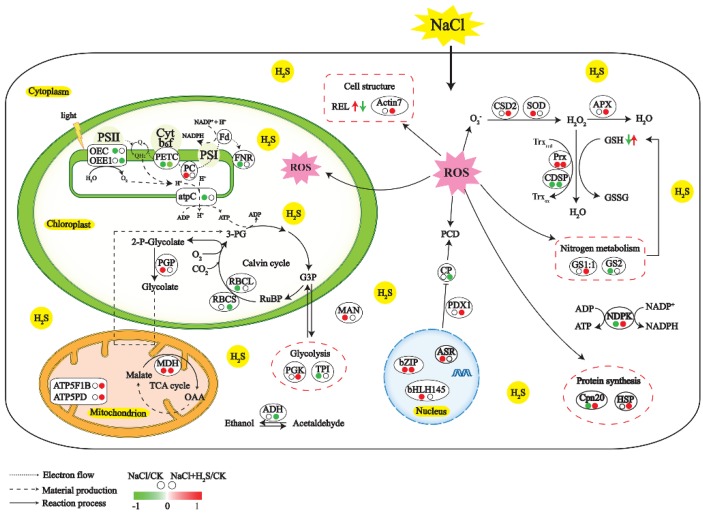
The proposed regulatory networks of H_2_S on the salt tolerance of *K. obovata* seedling leaves. The left and right dots stand for the NaCl and NaCl and H_2_S treatments, respectively. The red color of dot indicates the up-regulated change, the green color of dot indicates the down-regulated change, and the white color of dot indicates no change in comparison to the control. The left and right arrows stand for the NaCl treatment and the NaCl and H_2_S treatment, respectively. The red color arrow indicates physiological indexes of increased change, and the green color arrow indicates physiological indexes of decreased change. ADH: alcohol dehydrogenase; APX: cytosolic ascorbate peroxidase; Asr: abscisic acid stress ripening protein; atpC: ATP synthase CF1 epsilon subunit; ATP5F1B: ATP synthase beta subunit; ATP5PD: ATP synthase subunit d, mitochondrial; CP: cysteine proteinase; Cpn 20: 20 kDa chaperonin family protein; Cpn 60: TCP-1/cpn60 chaperonin family protein; CSD2: copper/zinc superoxide dismutase; CDSP: chloroplastic drought-induced stress protein; FLDH: NAD(P)-binding Rossmann-fold superfamily protein isoform 1; FNR: ferredoxin-NADP reductase; GS2: glutamine synthetase 2; GS1:1: glutamine synthetase 1;1; HSP: heat-shock protein; MAN: Glycosyl hydrolase superfamily protein; MDH: malate dehydrogenase; NDPK: nucleoside diphosphate kinase; OEE1: oxygen-evolving enhancer protein; OEC: O2 evolving complex 33kD family protein; PC: plastocyanin; TPI: triosephosphate isomerase; PCD: programmed cell death; PDX1: pyridoxin biosynthesis; PETC: photosynthetic electron transfer C; PGK: phosphoglycerate kinase; PGP: phosphoglycolate phosphatase; Prx: 2-Cys peroxiredoxin; RBCL: ribulose-1,5-bisphosphate carboxylase large subunit; RBCS: RuBisCO activase small isoform precursor; and SOD: superoxide dismutase. The detailed information for each spot is shown in Table 1.

**Table 1 ijms-21-00118-t001:** Identification of differentially expressed proteins with greater than 2.0-fold change in the leaves of *K. obovata* seedlings under the salinity and NaHS treatments.

								Ratio ^9^	
Spot ^1^	Accession (gb) ^2^	Protein Name ^3^	Theoretical (Mr/pI) ^4^	Observed (Mr/pI) ^5^	Score ^6^	MP ^7^	Species ^8^	NaCl vs. CK	NaCl and H_2_S vs. CK
**Photosynthesis**									
1	gi|502131189	PREDICTED: plastocyanin, chloroplastic-like	17.13/5.04	14.05/4.37	118	1	*Cicer arietinum*	2.257 ± 0.193	0.841 ± 0.068
2	gi|226497434	Ferredoxin-NADP reductase, leaf isozyme	40.98/8.53	37.32/6.74	311	4	*Zea mays*	0.287 ± 0.055	0.686 ± 0.153
8	gi|363981020	Ribulose-1,5-bisphosphate carboxylase/oxygenase large subunit, partial (chloroplast)	50.67/5.87	19.87/5.19	116	7	*Cercidiphyllum japonicum*	0.086 ± 0.014	0.736 ± 0.084
10	gi|508707371	Photosynthetic electron transfer C	24.67/8.15	19.14/4.79	103	5	*Theobroma cacao*	0.100 ± 0.005	0.483 ± 0.068
12	gi|222853091	O_2_ evolving complex 33kD family protein	35.41/5.85	36.80/5.22	405	4	*Populus trichocarpa*	0.067 ± 0.006	0.518 ± 0.065
14	gi|527190719	Oxygen-evolving enhancer protein 1, chloroplastic	34.80/6.48	34.78/5.47	493	8	*Genlisea aurea*	1.213 ± 0.061	0.068 ± 0.012
36	gi|62733297	RuBisCO activase small isoform precursor	52.39/5.59	42.29/5.03	276	7	*Oryza sativa* Japonica Group	0.741 ± 0.157	0.079 ± 0.033
**Carbohydrate and energy metabolism**									
6	gi|508785499	Glycosyl hydrolase superfamily protein, putative	37.45/8.39	34.33/4.66	68	1	*Theobroma cacao*	2.072 ± 0.431	1.279 ± 0.188
7	gi|336041766	ATP synthase CF1 epsilon subunit	13.69/5.43	14.89/5.03	97	2	*Justicia americana*	0.323 ± 0.098	0.528 ± 0.060
9	gi|460411739	PREDICTED: ATP synthase subunit d, mitochondrial-like	19.78/5.33	18.95/5.04	119	4	*Solanum lycopersicum*	0.704 ± 0.059	3.630 ± 0.341
16	gi|508786769	Triosephosphate isomerase	27.50/5.54	23.96/6.35	164	7	*Theobroma cacao*	0.142 ± 0.038	0.605 ± 0.061
20	gi|223535342	Alcohol dehydrogenase, putative	41.61/8.61	37.23/5.88	513	6	*Ricinus communis*	0.643 ± 0.158	0.311 ± 0.064
22	gi|226503019	Malate dehydrogenase, cytoplasmic	35.85/5.76	40.48/6.78	250	7	*Zea mays*	3.659 ± 0.520	2.997 ± 0.469
24	gi|508715598	NAD(P)-binding Rossmann-fold superfamily protein isoform 1	36.50/9.29	28.24/5.53	268	4	*Theobroma cacao*	0.663 ± 0.131	0.469 ± 0.079
25	gi|502105712	Phosphoglycolate phosphatase	40.78/6.89	37.32/6.74	226	10	*Cicer arietinum*	2.400 ± 0.373	1.927 ± 0.395
29	gi|335059237	ATP synthase beta subunit	52.29/5.09	57.59/4.97	453	11	*Clerodendrum trichotomum*	1.591 ± 0.062	2.706 ± 0.178
30	gi|475549973	Nucleoside diphosphate kinase 1	17.03/6.85	16.16/6.40	107	5	*Aegilops tauschii*	0.419 ± 0.040	9.366 ± 1.258
35	gi|223547261	Phosphoglycerate kinase, putative	50.11/8.74	51.05/5.99	294	6	*Ricinus communis*	1.115 ± 0.256	3.040 ± 0.592
**Amino acid and protein metabolism**									
4	gi|355492134	Cysteine proteinase	53.52/8.12	40.03/4.48	96	5	*Medicago truncatula*	1.512 ± 0.206	0.162 ± 0.012
19	gi|223544592	Heat-shock protein, putative	17.81/5.93	16.81/5.69	129	6	*Ricinus communis*	0.734 ± 0.071	2.172 ± 0.799
26	gi|550336292	20 kDa chaperonin family protein	26.89/8.75	23.19/5.62	245	5	*Populus trichocarpa*	0.196 ± 0.092	2.934 ± 0.618
28	gi|508779629	TCP-1/cpn60 chaperonin family protein	65.05/5.57	56.09/5.27	714	15	*Theobroma cacao*	2.301 ± 0.319	1.562 ± 0.134
32	gi|508713595	Glutamine synthetase 2	61.64/8.38	44.25/5.98	63	3	*Theobroma cacao*	0.208 ± 0.032	1.068 ± 0.160
33	gi|332006826	Glutamine synthetase 1;1	39.32/5.28	40.86/5.71	241	7	*Arabidopsis thaliana*	1.294 ± 0.280	1.807 ± 0.246
**Hormone biosynthesis and transcription factor**									
13	gi|332008500	Transcription factor bHLH145	35.26/5.08	40.10/5.52	64	11	*Arabidopsis thaliana*	4.254 ± 0.498	1.032 ± 0.079
21	gi|508712975	Mitochondrial transcription termination factor family protein, putative isoform 1	51.01/9.32	44.71/6.38	56	11	*Theobroma cacao*	0.252 ± 0.032	0.393 ± 0.063
27	gi|355514936	BZIP transcription factor	24.12/8.65	23.37/5.88	60	9	*Medicago truncatula*	2.478 ± 0.158	3.198 ± 0.157
31	gi|355481146	Abscisic acid stress ripening protein	27.33/5.15	39.71/5.9	207	2	*Medicago truncatula*	2.024 ± 0.207	1.567 ± 0.182
**Stress response proteins**									
5	gi|502112102	PREDICTED: 2-Cys peroxiredoxin BAS1-like, chloroplastic-like isoform X2	29.14/6.12	21.46/4.96	361	6	*Cucumis sativus*	2.446 ± 0.616	2.257 ± 0.502
11	gi|222867611	Pyridoxin biosynthesis PDX1-like protein 3	33.31/6.55	15.98/5.29	60	8	*Populus trichocarpa*	0.886 ± 0.266	2.230 ± 0.521
17	gi|409900374	Copper/zinc superoxide dismutase CSD2A-1	23.29/6.12	19.07/5.74	393	4	*Musa acuminata*	1.184 ± 0.167	2.006 ± 0.334
18	gi|414866828	Superoxide dismutase 4	15.65/5.10	15.02/5.80	77	3	*Zea mays*	2.510 ± 0.688	0.988 ± 0.100
23	gi|508776520	Actin 7 isoform 1	41.80/5.31	43.65/5.52	342	9	*Theobroma cacao*	0.827 ± 0.210	3.400 ± 0.725
34	gi|508723241	Chloroplastic drought-induced stress protein of 32 kD	40.91/7.66	27.45/5.77	213	6	*Theobroma cacao*	0.237 ± 0.078	0.210 ± 0.058
37	gi|498923199	Cytosolic ascorbate peroxidase	27.09/5.52	32.52/6.66	234	3	*Arachis hypogaea*	0.734 ± 0.235	2.172 ± 0.334
**Unknown protein**									
3	gi|548848586	Hypothetical protein AMTR_s00157p00064680	54.99/6.91	28.30/4.86	68	1	*Amborella trichopoda*	0.120 ± 0.011	0.759 ± 0.362
15	gi|462410037	Hypothetical protein PRUPE_ppa019045mg	25.73/5.65	29.45/6.01	150	5	*Prunus persica*	0.488 ± 0.109	0.669 ± 0.116

^1^ The spot number corresponds to the number listed in the table. ^2^ Database accession numbers (gb) according to NCBInr. ^3^ The names of proteins were identified by LC-MALDI-TOF/TOF. ^4^ Theoretical mass (kDa) and pI of identified proteins. Theoretical values were retrieved from the NCBInr database. ^5^ Experimental mass (kDa) and pI of identified proteins. Experimental values were calculated by using the PDQuest software (Version 8.0, Bio-Rad, Hercules, CA, USA) and standard molecular mass. ^6^ The Mascot searched score against the database NCBInr. ^7^ Number of matched peptide fragments. ^8^ The species which has the high homology of the identified protein. ^9^ Ratio between the different treatments. NaCl vs. CK means 400 mM of NaCl vs. control; NaCl and H_2_S vs. CK means 400 mM of NaCl and 200 μM of NaHS vs. control.

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
