# Peer review of "Comparative Proteomic Analysis Reveals the Regulatory Effects of H2S on Salt Tolerance of Mangrove Plant Kandelia obovata"

_ijms, 2019, doi:10.3390/ijms21010118_

Round 1
Reviewer 1 Report
In this manuscript Liu et al study the effects of H2S on salt tolerance of Kandelia obovata. The manuscript is very well written and provides a very interesting proteomic analysis that correlates well with the physiological data provided in the manuscript. Overall, I just have a couple of very small comments:
Comments:
-In Figure 1 the meaning of CK and NaCl is not explained in the figure legend or in the text.
-The data presented in Figures 3 and 4 is very important and I think it is not explained well in detail. Could the authors elaborate more about the differences in protein accumulation between the different treatments?
-Furthermore, is the alleviation of photosynthesis inhibition mediated by H2S represented at the molecular level? This is only discussed in the discussion part but I feel that it should also be mentioned in the body of the results part.
Typos:
-Line 320: “response to reponse”
-Line 417: “which involved”
-Line 417:” were found positivity modulated”
-Line 495: “in salt tolerance of K.”
Author Response
Response to Reviewer 1 Comments
Comments:
Point 1: In Figure 1 the meaning of CK and NaCl is not explained in the figure legend or in the text.
Response 1:
Original contents:
Figure 1. Effects of various concentrations of NaHS on leaf photosynthetic characteristics of K. obovata seedlings treated by high salinity. (A) leaf dry weight, (B) total chlorophyll contents, (C) net photosynthetic rate (Pn), (D) intercellular CO2 concentration (Ci), (E) stomatal conductance (Gs) and (F) transpiration rate (Tr). Seedlings were treated with different concentrations of NaHS (0, 50, 100, 200, 350 and 500 μM) together with 400 mM NaCl for 7 days. Values are means ± SE of three independent experiments with at least three replicates for each. Bars with different letters are significantly different at P < 0.05 according to Duncan's multiple range test.
Modified:
Figure 1. Effects of various concentrations of NaHS on leaf photosynthetic characteristics of K. obovata seedlings treated by high salinity. (A) leaf dry weight, (B) total chlorophyll contents, (C) net photosynthetic rate (Pn), (D) intercellular CO2 concentration (Ci), (E) stomatal conductance (Gs) and (F) transpiration rate (Tr). Seedlings were treated with different concentrations of NaHS (0, 50, 100, 200, 350 and 500 μM) together with 400 mM NaCl for 7 days. The sample without 400 mM NaCl was regarded as CK and the sample with 400 mM NaCl was regarded as NaCl. Values are means ± SE of three independent experiments with at least three replicates for each. Bars with different letters are significantly different at P < 0.05 according to Duncan's multiple range test.
Point 2: The data presented in Figures 3 and 4 is very important and I think it is not explained well in detail. Could the authors elaborate more about the differences in protein accumulation between the different treatments?
Response 2:
Original contents:
The 2-DE proteomic approach was employed to analyze the protein expression profile to investigate the mechanism of H2S-mediated salt tolerance of K. obovata. Three biological replicates were performed with similar results. ccording to the image analysis by PDQuest software, approximately 450 protein spots were reproducibly resolved on each gel. A total of 37 spots with significant changes (higher than 2.0-fold, P < 0.05) were identified as DEPs.
We provide 37 DEPs information in Table 1. A comprehensive overview of expression patterns of 37 identified proteins is shown in Figure 4A. They were divided into five functional categories, including carbohydrate and energy metabolism (29.73%), photosynthesis (18.92%), stress response proteins and cell structure (18.92%), amino acid and protein metabolism (16.22%), hormone biosynthesis and transcription factor (10.81%) by the biological function, which is in the UniProt database (Figure 4B). Comparing with NaCl vs CK group, there were more up-regulated DEPs in NaCl+H2S vs CK group. The up-regulated DEPs were mostly divided into carbohydrate and energy metabolism, and stress response proteins (Figure 4B).
Modified:
The 2-DE proteomic approach was employed to analyze the protein expression profile to investigate the mechanism of H2S-mediated salt tolerance of K. obovata. Three biological replicates were performed with similar results. ccording to the image analysis by PDQuest software, approximately 450 protein spots were reproducibly resolved on each gel. The protein spots were detected in the isoelectric point (pI) range from 4 to 7 and the range of molecular weight (MW) was 14.4 to 66.2 kDa. A total of 37 spots with significant changes (higher than 2.0-fold, P < 0.05) were identified as DEPs and presented in representative image (Figure 3A). Close-up images from four protein spots were displayed in Figure 3B.
We provide 37 DEPs information in Table 1. The difference between two groups (NaCl vs CK and NaCl+H2S vs CK) has statistically significant. A comprehensive overview of expression patterns of 37 identified proteins is shown in Figure 4A. They were divided into five functional categories, including carbohydrate and energy metabolism (29.73%), photosynthesis (18.92%), stress response proteins and cell structure (18.92%), amino acid and protein metabolism (16.22%), hormone biosynthesis and transcription factor (10.81%) by the biological function, which is in the UniProt database (Figure 4B). There were fifteen proteins significantly changed under NaCl treatment; specifically, the abundances of seven proteins were increased and eight proteins were decreased. Under NaCl+H2S treatment, fourteen proteins were significantly changed, the abundances of nine proteins were increased and five proteins were decreased. There were six proteins were affected by both NaCl and NaCl+H2S treatment (Figure 4A). Comparing with NaCl vs CK group, there were more up-regulated DEPs in NaCl+H2S vs CK group. These up-regulated DEPs were mostly divided into carbohydrate and energy metabolism, and stress response proteins (Figure 4B). There were less down-regulated DEPs divided into carbohydrate and energy metabolism, and photosynthesis under NaCl+H2S treatment (Figure 4B).
Point 3: Furthermore, is the alleviation of photosynthesis inhibition mediated by H2S represented at the molecular level? This is only discussed in the discussion part but I feel that it should also be mentioned in the body of the results part.
Response 3:
We add a description to the paragraph.
Comparing with NaCl vs CK group, there were more up-regulated DEPs in NaCl+H2S vs CK group. These up-regulated DEPs were mostly divided into carbohydrate and energy metabolism, and stress response proteins (Figure 4B). There were less down-regulated DEPs divided into carbohydrate and energy metabolism, and photosynthesis under NaCl+H2S treatment (Figure 4B).
Typos:
Point 1: Line 320: “response to reponse” abiotic stimulus,
Response 1: …… response to abiotic stimulus, ……
Point 2: Line 417: eat-shock protein (HSP, spot 19) and 20 kDa chaperonin family protein (Cpn 20, spot 26), “which involved” in protein synthesis, were found positivity modulated by NaHS addition.
Response 2: Heat-shock protein (HSP, spot 19) and 20 kDa chaperonin family protein (Cpn 20, spot 26) were involved in protein synthesis, which were positivity modulated by NaHS addition.
Point 3: Line 417:” were found positivity modulated”
Response 3: Heat-shock protein (HSP, spot 19) and 20 kDa chaperonin family protein (Cpn 20, spot 26) were involved in protein synthesis, which were positivity modulated by NaHS addition.
Point 4: Line 495: “in salt tolerance of K.”
Response 4: in salt tolerance of K. obovate.

Reviewer 2 Report
In the manuscript "Comparative proteomic analysis reveals the regulatory effects of H2S on salt tolerance of mangrove plant Kandelia obovata" by Yi-Ling Liu and colleagues, the authors compare the effect of NaCl and NaCl+H2S treatments on salt tolerance of Kandelia obovata combining physiological and proteomic analysis.
The paper falls within the scope of the journal and the presented results are interesting since the authors identify some metabolic pathways affected by H2S which alleviate NaCl symptoms.
Nevertheless, the English language needs an extensive editing and this is the main condition to make the paper suitable for publication in IJMS.
Moreover, sometimes the conclusions coming from the results are misleading since the authors discuss results without statistic relevance. In all these cases the text should be opportunely changed.
Hence, I suggest the manuscript for publication only after a thorough revision according to the hints in the attached file.

Author Response
Response to Reviewer 2 Comments
|
Page 2 |
|||||||||||||||||||||||||||||||||||||||||||||||||||||||||||||||||||||||||||||||||||||||||||||||||||||||||||||||||||||||||||||||||||||||||||||||||||||||||||||||||||||||||||||||||||||||||||||||||||||||||||||||||||||||||||||||||||||||||||||||||||||||||||||||||||||||||||||||
|
Point 1: |
The sentence is not clear, please rephrase it. |
||||||||||||||||||||||||||||||||||||||||||||||||||||||||||||||||||||||||||||||||||||||||||||||||||||||||||||||||||||||||||||||||||||||||||||||||||||||||||||||||||||||||||||||||||||||||||||||||||||||||||||||||||||||||||||||||||||||||||||||||||||||||||||||||||||||||||||||
|
Response 1: |
Original contents: Line 45-46: “During anaerobic decomposition, sulfate reduction in sediment was triggering, and H2S was consequent producing, which annual mean is 768 ± 240 μg·S·m-2 ·d-1.” Modified: The mangrove forest served as a perpetual source of H2S and the annual mean emission of H2S was 768 ± 240 μg·S·m-2·d-1. |
||||||||||||||||||||||||||||||||||||||||||||||||||||||||||||||||||||||||||||||||||||||||||||||||||||||||||||||||||||||||||||||||||||||||||||||||||||||||||||||||||||||||||||||||||||||||||||||||||||||||||||||||||||||||||||||||||||||||||||||||||||||||||||||||||||||||||||||
|
Point 2: |
Change as: D/L-CD |
||||||||||||||||||||||||||||||||||||||||||||||||||||||||||||||||||||||||||||||||||||||||||||||||||||||||||||||||||||||||||||||||||||||||||||||||||||||||||||||||||||||||||||||||||||||||||||||||||||||||||||||||||||||||||||||||||||||||||||||||||||||||||||||||||||||||||||||
|
Response 2: |
Original contents: Line 53: D/L-cysteine desulfhydrase (L/D-CD) Modified: D/L-cysteine desulfhydrase (D/L-CD) |
||||||||||||||||||||||||||||||||||||||||||||||||||||||||||||||||||||||||||||||||||||||||||||||||||||||||||||||||||||||||||||||||||||||||||||||||||||||||||||||||||||||||||||||||||||||||||||||||||||||||||||||||||||||||||||||||||||||||||||||||||||||||||||||||||||||||||||||
|
Point 3: |
Delete "re-growth ability ". It is a repetition |
||||||||||||||||||||||||||||||||||||||||||||||||||||||||||||||||||||||||||||||||||||||||||||||||||||||||||||||||||||||||||||||||||||||||||||||||||||||||||||||||||||||||||||||||||||||||||||||||||||||||||||||||||||||||||||||||||||||||||||||||||||||||||||||||||||||||||||||
|
Response 3: |
Original contents: Line 55: ……tobacco suspension-cultured cells re-growth ability after heat stress…… Modified: ……tobacco suspension-cultured cells after heat stress …… |
||||||||||||||||||||||||||||||||||||||||||||||||||||||||||||||||||||||||||||||||||||||||||||||||||||||||||||||||||||||||||||||||||||||||||||||||||||||||||||||||||||||||||||||||||||||||||||||||||||||||||||||||||||||||||||||||||||||||||||||||||||||||||||||||||||||||||||||
|
Point 4: |
The sentence seems uncomplete, please rephrase |
||||||||||||||||||||||||||||||||||||||||||||||||||||||||||||||||||||||||||||||||||||||||||||||||||||||||||||||||||||||||||||||||||||||||||||||||||||||||||||||||||||||||||||||||||||||||||||||||||||||||||||||||||||||||||||||||||||||||||||||||||||||||||||||||||||||||||||||
|
Response 4: |
Original contents: Line 56: Salinity can detrimentally impact plant biomass production, which is associated with the salinity-induced. Modified: Salinity can detrimentally impact plant biomass production, which is associated with the salinity-induced a reduction in the amount of chlorophyll. |
||||||||||||||||||||||||||||||||||||||||||||||||||||||||||||||||||||||||||||||||||||||||||||||||||||||||||||||||||||||||||||||||||||||||||||||||||||||||||||||||||||||||||||||||||||||||||||||||||||||||||||||||||||||||||||||||||||||||||||||||||||||||||||||||||||||||||||||
|
Point 5: |
Change as: lipid peroxides or lipid peroxidation |
||||||||||||||||||||||||||||||||||||||||||||||||||||||||||||||||||||||||||||||||||||||||||||||||||||||||||||||||||||||||||||||||||||||||||||||||||||||||||||||||||||||||||||||||||||||||||||||||||||||||||||||||||||||||||||||||||||||||||||||||||||||||||||||||||||||||||||||
|
Response 5: |
Original contents: Line 64: lipid peroxide and hydrogen peroxide (H2O2) Modified: lipid peroxides and hydrogen peroxide (H2O2) |
||||||||||||||||||||||||||||||||||||||||||||||||||||||||||||||||||||||||||||||||||||||||||||||||||||||||||||||||||||||||||||||||||||||||||||||||||||||||||||||||||||||||||||||||||||||||||||||||||||||||||||||||||||||||||||||||||||||||||||||||||||||||||||||||||||||||||||||
|
Point 6: |
Change as: H2S-mediated α- and β-amylase in the endosperm |
||||||||||||||||||||||||||||||||||||||||||||||||||||||||||||||||||||||||||||||||||||||||||||||||||||||||||||||||||||||||||||||||||||||||||||||||||||||||||||||||||||||||||||||||||||||||||||||||||||||||||||||||||||||||||||||||||||||||||||||||||||||||||||||||||||||||||||||
|
Response 6: |
Original contents: Line 65: H2S-mediated α- and β-amylase in the endosperm Modified: H2S-mediated α- and β-amylase induction in the endosperm |
||||||||||||||||||||||||||||||||||||||||||||||||||||||||||||||||||||||||||||||||||||||||||||||||||||||||||||||||||||||||||||||||||||||||||||||||||||||||||||||||||||||||||||||||||||||||||||||||||||||||||||||||||||||||||||||||||||||||||||||||||||||||||||||||||||||||||||||
|
Point 7: |
Change as: roots pre-treatment with H2S induced an increase in..... |
||||||||||||||||||||||||||||||||||||||||||||||||||||||||||||||||||||||||||||||||||||||||||||||||||||||||||||||||||||||||||||||||||||||||||||||||||||||||||||||||||||||||||||||||||||||||||||||||||||||||||||||||||||||||||||||||||||||||||||||||||||||||||||||||||||||||||||||
|
Response 7: |
Original contents: Line 67: pre-treatment roots with H2S have a distinct increase in stomatal conductance, Modified: root pre-treatment with H2S induced an increase in stomatal conductance, |
||||||||||||||||||||||||||||||||||||||||||||||||||||||||||||||||||||||||||||||||||||||||||||||||||||||||||||||||||||||||||||||||||||||||||||||||||||||||||||||||||||||||||||||||||||||||||||||||||||||||||||||||||||||||||||||||||||||||||||||||||||||||||||||||||||||||||||||
|
Page 3 |
|||||||||||||||||||||||||||||||||||||||||||||||||||||||||||||||||||||||||||||||||||||||||||||||||||||||||||||||||||||||||||||||||||||||||||||||||||||||||||||||||||||||||||||||||||||||||||||||||||||||||||||||||||||||||||||||||||||||||||||||||||||||||||||||||||||||||||||||
|
Point 8: |
“ verb is missing, such as counteract” |
||||||||||||||||||||||||||||||||||||||||||||||||||||||||||||||||||||||||||||||||||||||||||||||||||||||||||||||||||||||||||||||||||||||||||||||||||||||||||||||||||||||||||||||||||||||||||||||||||||||||||||||||||||||||||||||||||||||||||||||||||||||||||||||||||||||||||||||
|
Response 8: |
Original contents: Line 98: After 7-days treatment, photosynthesis, chlorophyll content and dry weight were measured to determine the optimal NaHS concentration that growth inhibition induced by NaCl. Modified: After 7-days treatment, photosynthesis, chlorophyll content and dry weight were measured to determine the optimal concentration of NaHS for the salinity-induced growth inhibition. |
||||||||||||||||||||||||||||||||||||||||||||||||||||||||||||||||||||||||||||||||||||||||||||||||||||||||||||||||||||||||||||||||||||||||||||||||||||||||||||||||||||||||||||||||||||||||||||||||||||||||||||||||||||||||||||||||||||||||||||||||||||||||||||||||||||||||||||||
|
Point 9: |
The last sentence is not clear, please rephrase |
||||||||||||||||||||||||||||||||||||||||||||||||||||||||||||||||||||||||||||||||||||||||||||||||||||||||||||||||||||||||||||||||||||||||||||||||||||||||||||||||||||||||||||||||||||||||||||||||||||||||||||||||||||||||||||||||||||||||||||||||||||||||||||||||||||||||||||||
|
Response 9: |
Original contents: Line 103: Leaves were harvested after 7 days treatments for chlorophyll and endogenous H2S content measurement to distinguish the actual role of H2S/HS- from the other possible compounds derived from NaHS decomposition. Modified: After 7 days treatments, the content of chlorophyll and endogenous H2S were measured to distinguish the actual role of H2S/HS- from the other possible compounds, which was derived from NaHS decomposition. |
||||||||||||||||||||||||||||||||||||||||||||||||||||||||||||||||||||||||||||||||||||||||||||||||||||||||||||||||||||||||||||||||||||||||||||||||||||||||||||||||||||||||||||||||||||||||||||||||||||||||||||||||||||||||||||||||||||||||||||||||||||||||||||||||||||||||||||||
|
Point 10: |
Actually, you measured TBARS content. Thus, you should name this parameter as TBARS rather than MDA, since assay is not specific to MDA. MDA is a lipid peroxidation product, which can react with TBA, but the assay actually measures all TBA reactive substances (hence, TBARS). Although, MDA is widely used in the literature, actually it is misleading.” |
||||||||||||||||||||||||||||||||||||||||||||||||||||||||||||||||||||||||||||||||||||||||||||||||||||||||||||||||||||||||||||||||||||||||||||||||||||||||||||||||||||||||||||||||||||||||||||||||||||||||||||||||||||||||||||||||||||||||||||||||||||||||||||||||||||||||||||||
|
Response 10: |
Original contents: Line 129: Lipid peroxidation in terms of malondialdehyde (MDA) content was measured according to Yan et al (2010) Modified: Lipid peroxidation was determined by measuring thiobarbituric acid reactive substances (TBARS) according to Yan et al (2010). |
||||||||||||||||||||||||||||||||||||||||||||||||||||||||||||||||||||||||||||||||||||||||||||||||||||||||||||||||||||||||||||||||||||||||||||||||||||||||||||||||||||||||||||||||||||||||||||||||||||||||||||||||||||||||||||||||||||||||||||||||||||||||||||||||||||||||||||||
|
Page 4-5 |
|||||||||||||||||||||||||||||||||||||||||||||||||||||||||||||||||||||||||||||||||||||||||||||||||||||||||||||||||||||||||||||||||||||||||||||||||||||||||||||||||||||||||||||||||||||||||||||||||||||||||||||||||||||||||||||||||||||||||||||||||||||||||||||||||||||||||||||||
|
Point 11: |
Change as: containing |
||||||||||||||||||||||||||||||||||||||||||||||||||||||||||||||||||||||||||||||||||||||||||||||||||||||||||||||||||||||||||||||||||||||||||||||||||||||||||||||||||||||||||||||||||||||||||||||||||||||||||||||||||||||||||||||||||||||||||||||||||||||||||||||||||||||||||||||
|
Response 11: |
Original contents: Line 150: (contain 1 mg protein sample) Modified: (containing 1 mg protein sample) |
||||||||||||||||||||||||||||||||||||||||||||||||||||||||||||||||||||||||||||||||||||||||||||||||||||||||||||||||||||||||||||||||||||||||||||||||||||||||||||||||||||||||||||||||||||||||||||||||||||||||||||||||||||||||||||||||||||||||||||||||||||||||||||||||||||||||||||||
|
Point 12: |
Please rephrase the last sentence |
||||||||||||||||||||||||||||||||||||||||||||||||||||||||||||||||||||||||||||||||||||||||||||||||||||||||||||||||||||||||||||||||||||||||||||||||||||||||||||||||||||||||||||||||||||||||||||||||||||||||||||||||||||||||||||||||||||||||||||||||||||||||||||||||||||||||||||||
|
Response 12: |
Original contents: Line 153: Each experiment was repeated three times. Modified: Each treatment was performed for three biological replications for quantitative analysis. |
||||||||||||||||||||||||||||||||||||||||||||||||||||||||||||||||||||||||||||||||||||||||||||||||||||||||||||||||||||||||||||||||||||||||||||||||||||||||||||||||||||||||||||||||||||||||||||||||||||||||||||||||||||||||||||||||||||||||||||||||||||||||||||||||||||||||||||||
|
Point 13: |
Please rephrase the last sentence |
||||||||||||||||||||||||||||||||||||||||||||||||||||||||||||||||||||||||||||||||||||||||||||||||||||||||||||||||||||||||||||||||||||||||||||||||||||||||||||||||||||||||||||||||||||||||||||||||||||||||||||||||||||||||||||||||||||||||||||||||||||||||||||||||||||||||||||||
|
Response 13: |
Original contents: Line 157: Afterward, those proteins which compared to the control gels by over 2.0-fold changes were selected for the next experiment. Modified: Afterward, differentially expressed proteins (DEPs) were obtained by pairwise comparison with a fold change ≥ 2.0 and a two-tailed Student’s t-test (P < 0.05). |
||||||||||||||||||||||||||||||||||||||||||||||||||||||||||||||||||||||||||||||||||||||||||||||||||||||||||||||||||||||||||||||||||||||||||||||||||||||||||||||||||||||||||||||||||||||||||||||||||||||||||||||||||||||||||||||||||||||||||||||||||||||||||||||||||||||||||||||
|
Point 14: |
“it is better to specific the time of incubation” |
||||||||||||||||||||||||||||||||||||||||||||||||||||||||||||||||||||||||||||||||||||||||||||||||||||||||||||||||||||||||||||||||||||||||||||||||||||||||||||||||||||||||||||||||||||||||||||||||||||||||||||||||||||||||||||||||||||||||||||||||||||||||||||||||||||||||||||||
|
Response 14: |
Original contents: Line 161: ……and acetonitrile for at least four times until the color of CBB was removed. Modified: ……and acetonitrile for four times to remove the color of CBB. |
||||||||||||||||||||||||||||||||||||||||||||||||||||||||||||||||||||||||||||||||||||||||||||||||||||||||||||||||||||||||||||||||||||||||||||||||||||||||||||||||||||||||||||||||||||||||||||||||||||||||||||||||||||||||||||||||||||||||||||||||||||||||||||||||||||||||||||||
|
Point 15: |
Change as: identification |
||||||||||||||||||||||||||||||||||||||||||||||||||||||||||||||||||||||||||||||||||||||||||||||||||||||||||||||||||||||||||||||||||||||||||||||||||||||||||||||||||||||||||||||||||||||||||||||||||||||||||||||||||||||||||||||||||||||||||||||||||||||||||||||||||||||||||||||
|
Response 15: |
Original contents: Line 169: ……higher than 95% were considered to be an identifications. Modified: ……higher than 95% were considered to be an identification. |
||||||||||||||||||||||||||||||||||||||||||||||||||||||||||||||||||||||||||||||||||||||||||||||||||||||||||||||||||||||||||||||||||||||||||||||||||||||||||||||||||||||||||||||||||||||||||||||||||||||||||||||||||||||||||||||||||||||||||||||||||||||||||||||||||||||||||||||
|
Point 16: |
Change as: are shown |
||||||||||||||||||||||||||||||||||||||||||||||||||||||||||||||||||||||||||||||||||||||||||||||||||||||||||||||||||||||||||||||||||||||||||||||||||||||||||||||||||||||||||||||||||||||||||||||||||||||||||||||||||||||||||||||||||||||||||||||||||||||||||||||||||||||||||||||
|
Response 16: |
Original contents: Line 187: The primers used for real-time PCR were shown…… Modified: The primers used for real-time PCR are shown…… |
||||||||||||||||||||||||||||||||||||||||||||||||||||||||||||||||||||||||||||||||||||||||||||||||||||||||||||||||||||||||||||||||||||||||||||||||||||||||||||||||||||||||||||||||||||||||||||||||||||||||||||||||||||||||||||||||||||||||||||||||||||||||||||||||||||||||||||||
|
Point 17: |
Line 188: The authors should report in the table also the efficiency of the primers and the R squared of the standard curve |
||||||||||||||||||||||||||||||||||||||||||||||||||||||||||||||||||||||||||||||||||||||||||||||||||||||||||||||||||||||||||||||||||||||||||||||||||||||||||||||||||||||||||||||||||||||||||||||||||||||||||||||||||||||||||||||||||||||||||||||||||||||||||||||||||||||||||||||
|
Response 17: |
See response 19: |
||||||||||||||||||||||||||||||||||||||||||||||||||||||||||||||||||||||||||||||||||||||||||||||||||||||||||||||||||||||||||||||||||||||||||||||||||||||||||||||||||||||||||||||||||||||||||||||||||||||||||||||||||||||||||||||||||||||||||||||||||||||||||||||||||||||||||||||
|
Point 18: |
Change as: performe |
||||||||||||||||||||||||||||||||||||||||||||||||||||||||||||||||||||||||||||||||||||||||||||||||||||||||||||||||||||||||||||||||||||||||||||||||||||||||||||||||||||||||||||||||||||||||||||||||||||||||||||||||||||||||||||||||||||||||||||||||||||||||||||||||||||||||||||||
|
Response 18: |
Original contents: Line 189: Five independent biological replicates were used to performed…… Modified: Five independent biological replicates were used to performe…… |
||||||||||||||||||||||||||||||||||||||||||||||||||||||||||||||||||||||||||||||||||||||||||||||||||||||||||||||||||||||||||||||||||||||||||||||||||||||||||||||||||||||||||||||||||||||||||||||||||||||||||||||||||||||||||||||||||||||||||||||||||||||||||||||||||||||||||||||
|
Point 19: |
Line 190: without any correction for primers efficiency? |
||||||||||||||||||||||||||||||||||||||||||||||||||||||||||||||||||||||||||||||||||||||||||||||||||||||||||||||||||||||||||||||||||||||||||||||||||||||||||||||||||||||||||||||||||||||||||||||||||||||||||||||||||||||||||||||||||||||||||||||||||||||||||||||||||||||||||||||
|
Response 19: Supplementary Table S1. Sequences of forward and reverse primers used in qPCR for gene expression analysis in K. obovata leaves.
|
|||||||||||||||||||||||||||||||||||||||||||||||||||||||||||||||||||||||||||||||||||||||||||||||||||||||||||||||||||||||||||||||||||||||||||||||||||||||||||||||||||||||||||||||||||||||||||||||||||||||||||||||||||||||||||||||||||||||||||||||||||||||||||||||||||||||||||||||
|
Point 20: |
Not adequated. On the other side it is better |
||||||||||||||||||||||||||||||||||||||||||||||||||||||||||||||||||||||||||||||||||||||||||||||||||||||||||||||||||||||||||||||||||||||||||||||||||||||||||||||||||||||||||||||||||||||||||||||||||||||||||||||||||||||||||||||||||||||||||||||||||||||||||||||||||||||||||||||
|
Response 20: |
Original contents: Line 204: Meanwhile, a high concentration…… Modified: On the other side, a high concentration…… |
||||||||||||||||||||||||||||||||||||||||||||||||||||||||||||||||||||||||||||||||||||||||||||||||||||||||||||||||||||||||||||||||||||||||||||||||||||||||||||||||||||||||||||||||||||||||||||||||||||||||||||||||||||||||||||||||||||||||||||||||||||||||||||||||||||||||||||||
|
Point 21: |
The authors should discuss (in discussion section) the negative correlation between Ci and Pn. It is peculiar that the combined treatments show an opposite effect on Ci and Pn. |
||||||||||||||||||||||||||||||||||||||||||||||||||||||||||||||||||||||||||||||||||||||||||||||||||||||||||||||||||||||||||||||||||||||||||||||||||||||||||||||||||||||||||||||||||||||||||||||||||||||||||||||||||||||||||||||||||||||||||||||||||||||||||||||||||||||||||||||
|
Response 21: |
In line 374-378: The change in Ci was analyzed as an indicator of the dominant factor of stomatal or non-stomatal limitations to photosynthesis. Ci showed a reverse trend to Gs, which suggests that non-stomatal closure is the dominant limitation to photosynthesis (Flexas and Medrano 2002). In the present research, the decrease in Ci showed a reverse trend to Gs under various concentrations of NaHS, which indicated that non-stomatal limitations dominated. |
||||||||||||||||||||||||||||||||||||||||||||||||||||||||||||||||||||||||||||||||||||||||||||||||||||||||||||||||||||||||||||||||||||||||||||||||||||||||||||||||||||||||||||||||||||||||||||||||||||||||||||||||||||||||||||||||||||||||||||||||||||||||||||||||||||||||||||||
|
Page 6 |
|||||||||||||||||||||||||||||||||||||||||||||||||||||||||||||||||||||||||||||||||||||||||||||||||||||||||||||||||||||||||||||||||||||||||||||||||||||||||||||||||||||||||||||||||||||||||||||||||||||||||||||||||||||||||||||||||||||||||||||||||||||||||||||||||||||||||||||||
|
Point 22: |
Why in B and F panel, the authors did not start with letter A in the first column? |
||||||||||||||||||||||||||||||||||||||||||||||||||||||||||||||||||||||||||||||||||||||||||||||||||||||||||||||||||||||||||||||||||||||||||||||||||||||||||||||||||||||||||||||||||||||||||||||||||||||||||||||||||||||||||||||||||||||||||||||||||||||||||||||||||||||||||||||
|
Response 22: |
We analyzed the difference between the various concentration of H2S with or without NaCl. Nevertheless, we didn't analyze the difference between NaCl treatment and NaCl+H2S treatment at the same H2S concentration. The level of physiological indexes was not highest in the first column of B and F panel, so we did not start with letter A in the first column of B and F panel. |
||||||||||||||||||||||||||||||||||||||||||||||||||||||||||||||||||||||||||||||||||||||||||||||||||||||||||||||||||||||||||||||||||||||||||||||||||||||||||||||||||||||||||||||||||||||||||||||||||||||||||||||||||||||||||||||||||||||||||||||||||||||||||||||||||||||||||||||
|
Point 23: |
Which ones? |
||||||||||||||||||||||||||||||||||||||||||||||||||||||||||||||||||||||||||||||||||||||||||||||||||||||||||||||||||||||||||||||||||||||||||||||||||||||||||||||||||||||||||||||||||||||||||||||||||||||||||||||||||||||||||||||||||||||||||||||||||||||||||||||||||||||||||||||
|
Response 23: |
Original contents: Line 224-225: The research results showed that, except NaHS, the above chemicals failed to rescue NaCl-induced reduction of chlorophyll content only except NaHS (Supplementary Figure S2A). Modified: The research results showed that, the above chemicals (Na2SO4, NaHSO3, NaHSO4, CH3COONa) failed to rescue NaCl-induced reduction of chlorophyll content only except NaHS (Supplementary Figure S2A). Delete “except NaHS, ” |
||||||||||||||||||||||||||||||||||||||||||||||||||||||||||||||||||||||||||||||||||||||||||||||||||||||||||||||||||||||||||||||||||||||||||||||||||||||||||||||||||||||||||||||||||||||||||||||||||||||||||||||||||||||||||||||||||||||||||||||||||||||||||||||||||||||||||||||
|
Point 24: |
This sentence is totally unclear. Please reword it. |
||||||||||||||||||||||||||||||||||||||||||||||||||||||||||||||||||||||||||||||||||||||||||||||||||||||||||||||||||||||||||||||||||||||||||||||||||||||||||||||||||||||||||||||||||||||||||||||||||||||||||||||||||||||||||||||||||||||||||||||||||||||||||||||||||||||||||||||
|
Response 24: |
Original contents: Line 226-228: Meanwhile, leaf endogenous H2S content kept the high level at NaHS treatment but maintained a stable low level under the negative controls when compared with the experimental control group (Supplementary Figure S2B). Modified: Meanwhile, compared with the experimental control group (CK), the endogenous H2S content kept the high level at NaHS treatment but maintained a stable low level under other sulfur-containing compounds or sodium-containing compounds (Supplementary Figure S2B). |
||||||||||||||||||||||||||||||||||||||||||||||||||||||||||||||||||||||||||||||||||||||||||||||||||||||||||||||||||||||||||||||||||||||||||||||||||||||||||||||||||||||||||||||||||||||||||||||||||||||||||||||||||||||||||||||||||||||||||||||||||||||||||||||||||||||||||||||
|
Point 25: |
Change as: Cytomembranes are damaged by ROS induced by salinity |
||||||||||||||||||||||||||||||||||||||||||||||||||||||||||||||||||||||||||||||||||||||||||||||||||||||||||||||||||||||||||||||||||||||||||||||||||||||||||||||||||||||||||||||||||||||||||||||||||||||||||||||||||||||||||||||||||||||||||||||||||||||||||||||||||||||||||||||
|
Response 25: |
Original contents: Line 242: Salinity was damaged to cytomembranes by increasing ROS. Modified: Cytomembranes are damaged by ROS induced by salinity. |
||||||||||||||||||||||||||||||||||||||||||||||||||||||||||||||||||||||||||||||||||||||||||||||||||||||||||||||||||||||||||||||||||||||||||||||||||||||||||||||||||||||||||||||||||||||||||||||||||||||||||||||||||||||||||||||||||||||||||||||||||||||||||||||||||||||||||||||
|
Point 26: |
Please note that the decrease is not significant. Change the test accordingly to this evidence. |
||||||||||||||||||||||||||||||||||||||||||||||||||||||||||||||||||||||||||||||||||||||||||||||||||||||||||||||||||||||||||||||||||||||||||||||||||||||||||||||||||||||||||||||||||||||||||||||||||||||||||||||||||||||||||||||||||||||||||||||||||||||||||||||||||||||||||||||
|
Response 26: |
Original contents: Line 243: the content of H2O2 decreased from 5.36 μmol·g -1 to 4.79 μmol·g -1 Modified: the content of H2O2 was decreased, but didn’t significantly differ from that of NaCl treatment |
||||||||||||||||||||||||||||||||||||||||||||||||||||||||||||||||||||||||||||||||||||||||||||||||||||||||||||||||||||||||||||||||||||||||||||||||||||||||||||||||||||||||||||||||||||||||||||||||||||||||||||||||||||||||||||||||||||||||||||||||||||||||||||||||||||||||||||||
|
Page 7 |
|||||||||||||||||||||||||||||||||||||||||||||||||||||||||||||||||||||||||||||||||||||||||||||||||||||||||||||||||||||||||||||||||||||||||||||||||||||||||||||||||||||||||||||||||||||||||||||||||||||||||||||||||||||||||||||||||||||||||||||||||||||||||||||||||||||||||||||||
|
Point 27: |
Please correct accordingly to the comment in Methods section |
||||||||||||||||||||||||||||||||||||||||||||||||||||||||||||||||||||||||||||||||||||||||||||||||||||||||||||||||||||||||||||||||||||||||||||||||||||||||||||||||||||||||||||||||||||||||||||||||||||||||||||||||||||||||||||||||||||||||||||||||||||||||||||||||||||||||||||||
|
Response 27: |
Original contents: Leaf REL and the content of MDA were measured to estimate the protective role of H2S on the stability of membrane. Modified: Line 245: Leaf REL and the TBARS accumulation were measured to estimate the protective role of H2S on the stability of membrane. We have modified the figure B. |
||||||||||||||||||||||||||||||||||||||||||||||||||||||||||||||||||||||||||||||||||||||||||||||||||||||||||||||||||||||||||||||||||||||||||||||||||||||||||||||||||||||||||||||||||||||||||||||||||||||||||||||||||||||||||||||||||||||||||||||||||||||||||||||||||||||||||||||
|
Point 28: |
The y-axis has a different scale in the figure. Moreover, the difference between NaCl and NaCl+H2S treatments is not statistically significant. Please change the text accordingly to this evidence |
||||||||||||||||||||||||||||||||||||||||||||||||||||||||||||||||||||||||||||||||||||||||||||||||||||||||||||||||||||||||||||||||||||||||||||||||||||||||||||||||||||||||||||||||||||||||||||||||||||||||||||||||||||||||||||||||||||||||||||||||||||||||||||||||||||||||||||||
|
Response 28: |
Original contents: Line 246: Compared with NaCl treatment, the content of MDA was decreased from 0.71 μmol·g -1 ·FW to around 0.58 μmol·g -1 ·FW by exogenous H2S addition (Figure 2B). Modified: The TBARS levels were decreased by exogenous H2S addition, but didn’t significantly differ from that of NaCl treatment (Figure 2B). |
||||||||||||||||||||||||||||||||||||||||||||||||||||||||||||||||||||||||||||||||||||||||||||||||||||||||||||||||||||||||||||||||||||||||||||||||||||||||||||||||||||||||||||||||||||||||||||||||||||||||||||||||||||||||||||||||||||||||||||||||||||||||||||||||||||||||||||||
|
Point 29: |
Note that the increase is not statistically significant in both cases. Please change the text accordingly to this evidence |
||||||||||||||||||||||||||||||||||||||||||||||||||||||||||||||||||||||||||||||||||||||||||||||||||||||||||||||||||||||||||||||||||||||||||||||||||||||||||||||||||||||||||||||||||||||||||||||||||||||||||||||||||||||||||||||||||||||||||||||||||||||||||||||||||||||||||||||
|
Response 29: |
Original contents: Line 253: APX activity increased by 6.1 U·g -1 ·min -1 (Figure 2E), while the activity of SOD increased by about 315.3 U·g -1 ·min -1 (Figure 2F) in comparison with the NaCl treatment. Modified: the activity of APX and SOD were increased, but didn’t significantly differ from that of NaCl treatment (Figure 2E, F). |
||||||||||||||||||||||||||||||||||||||||||||||||||||||||||||||||||||||||||||||||||||||||||||||||||||||||||||||||||||||||||||||||||||||||||||||||||||||||||||||||||||||||||||||||||||||||||||||||||||||||||||||||||||||||||||||||||||||||||||||||||||||||||||||||||||||||||||||
|
Point 30: |
Add (REL) |
||||||||||||||||||||||||||||||||||||||||||||||||||||||||||||||||||||||||||||||||||||||||||||||||||||||||||||||||||||||||||||||||||||||||||||||||||||||||||||||||||||||||||||||||||||||||||||||||||||||||||||||||||||||||||||||||||||||||||||||||||||||||||||||||||||||||||||||
|
Response 30: |
Original contents: Line 258: (B) malondialdehyde (MDA) content, (C) electrolyte leakage percentage Modified: (B) thiobarbituric acid reactive substances (TBARS) accumulation, (C) electrolyte leakage percentage (REL) |
||||||||||||||||||||||||||||||||||||||||||||||||||||||||||||||||||||||||||||||||||||||||||||||||||||||||||||||||||||||||||||||||||||||||||||||||||||||||||||||||||||||||||||||||||||||||||||||||||||||||||||||||||||||||||||||||||||||||||||||||||||||||||||||||||||||||||||||
|
Point 31: |
According to which test? |
||||||||||||||||||||||||||||||||||||||||||||||||||||||||||||||||||||||||||||||||||||||||||||||||||||||||||||||||||||||||||||||||||||||||||||||||||||||||||||||||||||||||||||||||||||||||||||||||||||||||||||||||||||||||||||||||||||||||||||||||||||||||||||||||||||||||||||||
|
Response 31: |
Original contents: Line 262: The columns labeled with different letters are significantly different at P < 0.05. Modified: The columns labeled with different letters are significantly different at P < 0.05 according to Duncan's multiple range test. |
||||||||||||||||||||||||||||||||||||||||||||||||||||||||||||||||||||||||||||||||||||||||||||||||||||||||||||||||||||||||||||||||||||||||||||||||||||||||||||||||||||||||||||||||||||||||||||||||||||||||||||||||||||||||||||||||||||||||||||||||||||||||||||||||||||||||||||||
|
Page 8-10 |
|||||||||||||||||||||||||||||||||||||||||||||||||||||||||||||||||||||||||||||||||||||||||||||||||||||||||||||||||||||||||||||||||||||||||||||||||||||||||||||||||||||||||||||||||||||||||||||||||||||||||||||||||||||||||||||||||||||||||||||||||||||||||||||||||||||||||||||||
|
Point 32: |
It would be interesting to know if the difference between the two ratios (i.e. NaCl vs CK and NaCl+H2S vs CK) is statistically significant. The authors should add this information. |
||||||||||||||||||||||||||||||||||||||||||||||||||||||||||||||||||||||||||||||||||||||||||||||||||||||||||||||||||||||||||||||||||||||||||||||||||||||||||||||||||||||||||||||||||||||||||||||||||||||||||||||||||||||||||||||||||||||||||||||||||||||||||||||||||||||||||||||
|
Response 32: |
We have added this information in the Table 1. |
||||||||||||||||||||||||||||||||||||||||||||||||||||||||||||||||||||||||||||||||||||||||||||||||||||||||||||||||||||||||||||||||||||||||||||||||||||||||||||||||||||||||||||||||||||||||||||||||||||||||||||||||||||||||||||||||||||||||||||||||||||||||||||||||||||||||||||||
|
|||||||||||||||||||||||||||||||||||||||||||||||||||||||||||||||||||||||||||||||||||||||||||||||||||||||||||||||||||||||||||||||||||||||||||||||||||||||||||||||||||||||||||||||||||||||||||||||||||||||||||||||||||||||||||||||||||||||||||||||||||||||||||||||||||||||||||||||
|
Point 33: |
This should be 7 This should be 6 |
||||||||||||||||||||||||||||||||||||||||||||||||||||||||||||||||||||||||||||||||||||||||||||||||||||||||||||||||||||||||||||||||||||||||||||||||||||||||||||||||||||||||||||||||||||||||||||||||||||||||||||||||||||||||||||||||||||||||||||||||||||||||||||||||||||||||||||||
|
Response 33: |
Original contents: Line 302: 6 Number of matched peptide fragments. Line 303: 7 The Mascot searched score against the database NCBInr. Modified: 6The Mascot searched score against the database NCBInr. 7 Number of matched peptide fragments. |
||||||||||||||||||||||||||||||||||||||||||||||||||||||||||||||||||||||||||||||||||||||||||||||||||||||||||||||||||||||||||||||||||||||||||||||||||||||||||||||||||||||||||||||||||||||||||||||||||||||||||||||||||||||||||||||||||||||||||||||||||||||||||||||||||||||||||||||
|
Point 34: |
Please, move Fig. 3 before Table1 |
||||||||||||||||||||||||||||||||||||||||||||||||||||||||||||||||||||||||||||||||||||||||||||||||||||||||||||||||||||||||||||||||||||||||||||||||||||||||||||||||||||||||||||||||||||||||||||||||||||||||||||||||||||||||||||||||||||||||||||||||||||||||||||||||||||||||||||||
|
Response 34: |
We have moved the Fig.3 before Table1 |
||||||||||||||||||||||||||||||||||||||||||||||||||||||||||||||||||||||||||||||||||||||||||||||||||||||||||||||||||||||||||||||||||||||||||||||||||||||||||||||||||||||||||||||||||||||||||||||||||||||||||||||||||||||||||||||||||||||||||||||||||||||||||||||||||||||||||||||
|
Point 35: |
The title of the figure should be changed, as the figure shows only the 2D analysis of the control sample; only few spots are compared between the three conditions. |
||||||||||||||||||||||||||||||||||||||||||||||||||||||||||||||||||||||||||||||||||||||||||||||||||||||||||||||||||||||||||||||||||||||||||||||||||||||||||||||||||||||||||||||||||||||||||||||||||||||||||||||||||||||||||||||||||||||||||||||||||||||||||||||||||||||||||||||
|
Response 35: |
Original contents: Line 291: Figure 3. 2-DE analysis of proteins extracted from the leaves of K. obovate exposed to 291 NaCl and combined NaCl and NaHS treatments. Modified: Figure 3. 2-DE analysis of proteins extracted from the leaves of K. obovata. |
||||||||||||||||||||||||||||||||||||||||||||||||||||||||||||||||||||||||||||||||||||||||||||||||||||||||||||||||||||||||||||||||||||||||||||||||||||||||||||||||||||||||||||||||||||||||||||||||||||||||||||||||||||||||||||||||||||||||||||||||||||||||||||||||||||||||||||||
|
Point 36: |
Please add "respectively" |
||||||||||||||||||||||||||||||||||||||||||||||||||||||||||||||||||||||||||||||||||||||||||||||||||||||||||||||||||||||||||||||||||||||||||||||||||||||||||||||||||||||||||||||||||||||||||||||||||||||||||||||||||||||||||||||||||||||||||||||||||||||||||||||||||||||||||||||
|
Response 36: |
Original contents: Line 298:……are indicated on the top and left of the gel Modified: ……are indicated on the top and left of the gel respectively. |
||||||||||||||||||||||||||||||||||||||||||||||||||||||||||||||||||||||||||||||||||||||||||||||||||||||||||||||||||||||||||||||||||||||||||||||||||||||||||||||||||||||||||||||||||||||||||||||||||||||||||||||||||||||||||||||||||||||||||||||||||||||||||||||||||||||||||||||
|
Page 11-12 |
|||||||||||||||||||||||||||||||||||||||||||||||||||||||||||||||||||||||||||||||||||||||||||||||||||||||||||||||||||||||||||||||||||||||||||||||||||||||||||||||||||||||||||||||||||||||||||||||||||||||||||||||||||||||||||||||||||||||||||||||||||||||||||||||||||||||||||||||
|
Point 37: |
Please add "was carried out" to the end of the sentence |
||||||||||||||||||||||||||||||||||||||||||||||||||||||||||||||||||||||||||||||||||||||||||||||||||||||||||||||||||||||||||||||||||||||||||||||||||||||||||||||||||||||||||||||||||||||||||||||||||||||||||||||||||||||||||||||||||||||||||||||||||||||||||||||||||||||||||||||
|
Response 37: |
Original contents: Line 310: ……find the corresponding homologous proteins from Arabidopsis thaliana Modified: ……find the corresponding homologous proteins from Arabidopsis thaliana was carried out. |
||||||||||||||||||||||||||||||||||||||||||||||||||||||||||||||||||||||||||||||||||||||||||||||||||||||||||||||||||||||||||||||||||||||||||||||||||||||||||||||||||||||||||||||||||||||||||||||||||||||||||||||||||||||||||||||||||||||||||||||||||||||||||||||||||||||||||||||
|
Point 38: |
The acronyms of the proteins should be written in full |
||||||||||||||||||||||||||||||||||||||||||||||||||||||||||||||||||||||||||||||||||||||||||||||||||||||||||||||||||||||||||||||||||||||||||||||||||||||||||||||||||||||||||||||||||||||||||||||||||||||||||||||||||||||||||||||||||||||||||||||||||||||||||||||||||||||||||||||
|
Response 38: |
Original contents: Line 316: The key node in module B is RBCL. Modified: The key node in module B is Ribulose-1,5-bisphosphate carboxylase large subunit (RBCL). |
||||||||||||||||||||||||||||||||||||||||||||||||||||||||||||||||||||||||||||||||||||||||||||||||||||||||||||||||||||||||||||||||||||||||||||||||||||||||||||||||||||||||||||||||||||||||||||||||||||||||||||||||||||||||||||||||||||||||||||||||||||||||||||||||||||||||||||||
|
Point 39: |
The acronyms of the proteins should be written in full in the legend |
||||||||||||||||||||||||||||||||||||||||||||||||||||||||||||||||||||||||||||||||||||||||||||||||||||||||||||||||||||||||||||||||||||||||||||||||||||||||||||||||||||||||||||||||||||||||||||||||||||||||||||||||||||||||||||||||||||||||||||||||||||||||||||||||||||||||||||||
|
Response 39: |
The acronyms of the proteins have written in full in the legend. |
||||||||||||||||||||||||||||||||||||||||||||||||||||||||||||||||||||||||||||||||||||||||||||||||||||||||||||||||||||||||||||||||||||||||||||||||||||||||||||||||||||||||||||||||||||||||||||||||||||||||||||||||||||||||||||||||||||||||||||||||||||||||||||||||||||||||||||||
|
Point 40: |
This is not true for bHLH145 and GS1. Please change the text accordingly to this evidence |
||||||||||||||||||||||||||||||||||||||||||||||||||||||||||||||||||||||||||||||||||||||||||||||||||||||||||||||||||||||||||||||||||||||||||||||||||||||||||||||||||||||||||||||||||||||||||||||||||||||||||||||||||||||||||||||||||||||||||||||||||||||||||||||||||||||||||||||
|
Response 40: |
Original contents: Line 339: As shown in Figure 6, all selected genes showed a similar pattern with their corresponding protein. Modified: As shown in Figure 6, four selected genes showed a similar pattern with their corresponding protein, except bHLH 145 and GS1;1. |
||||||||||||||||||||||||||||||||||||||||||||||||||||||||||||||||||||||||||||||||||||||||||||||||||||||||||||||||||||||||||||||||||||||||||||||||||||||||||||||||||||||||||||||||||||||||||||||||||||||||||||||||||||||||||||||||||||||||||||||||||||||||||||||||||||||||||||||
|
Page 13 |
|||||||||||||||||||||||||||||||||||||||||||||||||||||||||||||||||||||||||||||||||||||||||||||||||||||||||||||||||||||||||||||||||||||||||||||||||||||||||||||||||||||||||||||||||||||||||||||||||||||||||||||||||||||||||||||||||||||||||||||||||||||||||||||||||||||||||||||||
|
Point 41: |
In this section the authors should discuss the results, not present them. Please, change the text accordingly |
||||||||||||||||||||||||||||||||||||||||||||||||||||||||||||||||||||||||||||||||||||||||||||||||||||||||||||||||||||||||||||||||||||||||||||||||||||||||||||||||||||||||||||||||||||||||||||||||||||||||||||||||||||||||||||||||||||||||||||||||||||||||||||||||||||||||||||||
|
Response 41: |
Original contents: Line 346: NaHS, which in solution dissociates to Na+ and HS- that associates with H+ producing H2S, is widely used to examine biological effects of H2S (Hosoki et al. 1997). To discriminate the effects of NaHS from other sulfur- and sodium-containing compounds, solutions of Na2SO4, NaHSO3, NaHSO4, and CH3COONa were used instead of NaHS to ameliorate NaCl-induced seedling growth inhibition. In contrast to the results from NaHS, treatment with other compounds failed to induce endogenous H2S accumulation and to alleviate the reduction in chlorophyll content caused by NaCl exposure (Supplementary Figure S2). Accordingly, this study provides evidence that NaHS-associated responses are H2S-specific. Modified: We delete the sentences “To discriminate the effects of NaHS from other sulfur- and sodium-containing compounds, solutions of Na2SO4, NaHSO3, NaHSO4, and CH3COONa were used instead of NaHS to ameliorate NaCl-induced seedling growth inhibition.” We change the text“Accordingly, this study provides evidence that NaHS contributes to chlorophyll content recovery by HS-, rather than Na+. NaHS-associated responses are H2S-specific in K.obovata. ” |
||||||||||||||||||||||||||||||||||||||||||||||||||||||||||||||||||||||||||||||||||||||||||||||||||||||||||||||||||||||||||||||||||||||||||||||||||||||||||||||||||||||||||||||||||||||||||||||||||||||||||||||||||||||||||||||||||||||||||||||||||||||||||||||||||||||||||||||
|
Point 42: |
From Fig. 1 the results are significant only for photosynthesis, where NaHS addition (until 200 micromolar) seems to neutralize the effect of NaCl restoring the condition of control samples. For leaf dry weight and chlorophyll content, the addition of NaHS does not seem to influence the effect of NaCl addition for most of the samples. Please change the text accordingly with this evidence. |
||||||||||||||||||||||||||||||||||||||||||||||||||||||||||||||||||||||||||||||||||||||||||||||||||||||||||||||||||||||||||||||||||||||||||||||||||||||||||||||||||||||||||||||||||||||||||||||||||||||||||||||||||||||||||||||||||||||||||||||||||||||||||||||||||||||||||||||
|
Response 42: |
Original contents: Line 358: Low concentration of H2S (50-350 μM) increased the K. obovata leaf dry weight, chlorophyll content, and photosynthesis. In contrast, when the concentration of H2S reached 500 μM, the above indices all decreased (Figure 1). These findings are in agreement with Chen et al. (2011), who reported a similar decline in the biomass and photosynthesis of Spinacia oleracea after treatment with a high concentration of NaHS. Modified: Low concentration of H2S (50-350 μM) increased the K. obovata leaf photosynthesis. In contrast, when the concentration of H2S reached 500 μM, the indice decreased (Figure 1). These findings are in agreement with Chen et al. (2011), who reported a similar decline in the photosynthesis of Spinacia oleracea after treatment with a high concentration of NaHS. |
||||||||||||||||||||||||||||||||||||||||||||||||||||||||||||||||||||||||||||||||||||||||||||||||||||||||||||||||||||||||||||||||||||||||||||||||||||||||||||||||||||||||||||||||||||||||||||||||||||||||||||||||||||||||||||||||||||||||||||||||||||||||||||||||||||||||||||||
|
Point 43: |
Please change their with its |
||||||||||||||||||||||||||||||||||||||||||||||||||||||||||||||||||||||||||||||||||||||||||||||||||||||||||||||||||||||||||||||||||||||||||||||||||||||||||||||||||||||||||||||||||||||||||||||||||||||||||||||||||||||||||||||||||||||||||||||||||||||||||||||||||||||||||||||
|
Response 43: |
Original contents: Line 362:……enhance photosynthesis reaching their maximal values…… Modified: ……enhance photosynthesis reaching its maximal values…… |
||||||||||||||||||||||||||||||||||||||||||||||||||||||||||||||||||||||||||||||||||||||||||||||||||||||||||||||||||||||||||||||||||||||||||||||||||||||||||||||||||||||||||||||||||||||||||||||||||||||||||||||||||||||||||||||||||||||||||||||||||||||||||||||||||||||||||||||
|
Point 44: |
Actually, the ratio between NaCl+H2S and CK is 0.5, not 1; thus it is not correct to say that the accumulation of OEC came back to control level under NaCl+H2S treatment. Please change the text accordingly with this evidence |
||||||||||||||||||||||||||||||||||||||||||||||||||||||||||||||||||||||||||||||||||||||||||||||||||||||||||||||||||||||||||||||||||||||||||||||||||||||||||||||||||||||||||||||||||||||||||||||||||||||||||||||||||||||||||||||||||||||||||||||||||||||||||||||||||||||||||||||
|
Response 44: |
Original contents: Line 371: The accumulation of OEC came back to control level under NaCl+H2S treatment, …… Modified: Although the accumulation of OEC was down-regulated in two groups, the accumulation of OEC was up-regulated relative to NaCl treatment,…… |
||||||||||||||||||||||||||||||||||||||||||||||||||||||||||||||||||||||||||||||||||||||||||||||||||||||||||||||||||||||||||||||||||||||||||||||||||||||||||||||||||||||||||||||||||||||||||||||||||||||||||||||||||||||||||||||||||||||||||||||||||||||||||||||||||||||||||||||
|
Point 45: |
The last two sentences should be integrated. The authors write "The capacity of photosynthesis is associated with the level of RBCL/RBCS ratio in rice leaves ", but how should be this ratio (> or < 1)? Moreover, they write about the up-regulation of RCA. Where do they find this evidence? |
||||||||||||||||||||||||||||||||||||||||||||||||||||||||||||||||||||||||||||||||||||||||||||||||||||||||||||||||||||||||||||||||||||||||||||||||||||||||||||||||||||||||||||||||||||||||||||||||||||||||||||||||||||||||||||||||||||||||||||||||||||||||||||||||||||||||||||||
|
Response 45: |
Original contents: Line 386: The capacity of photosynthesis is associated with the level of RBCL/RBCS ratio in rice leaves (Wang et al. 2009). The up-regulation of RCA and RBCL/RBCS ratio …… Modified: The capacity of photosynthesis is associated with the level of RBCL/RBCS ratio and the higher photosynthetic rate exhibited the higher RBCL/RBCS ratio in rice leaves (Wang et al. 2009). The up-regulation of RBCL/RBCS ratio …… |
||||||||||||||||||||||||||||||||||||||||||||||||||||||||||||||||||||||||||||||||||||||||||||||||||||||||||||||||||||||||||||||||||||||||||||||||||||||||||||||||||||||||||||||||||||||||||||||||||||||||||||||||||||||||||||||||||||||||||||||||||||||||||||||||||||||||||||||
|
Page 14 |
|||||||||||||||||||||||||||||||||||||||||||||||||||||||||||||||||||||||||||||||||||||||||||||||||||||||||||||||||||||||||||||||||||||||||||||||||||||||||||||||||||||||||||||||||||||||||||||||||||||||||||||||||||||||||||||||||||||||||||||||||||||||||||||||||||||||||||||||
|
Point 46: |
This sentence is truncated; it misses the main verb |
||||||||||||||||||||||||||||||||||||||||||||||||||||||||||||||||||||||||||||||||||||||||||||||||||||||||||||||||||||||||||||||||||||||||||||||||||||||||||||||||||||||||||||||||||||||||||||||||||||||||||||||||||||||||||||||||||||||||||||||||||||||||||||||||||||||||||||||
|
Response 46: |
Original contents: Line 392: Phosphoglycolate phosphatase (PGP, spot 25), which is involved in photorespiration and required for giving carbon from 2-phosphoglycolate back into metabolism Modified: Phosphoglycolate phosphatase (PGP, spot 25) is involved in photorespiration, which is required for giving carbon from 2-phosphoglycolate back into metabolism. |
||||||||||||||||||||||||||||||||||||||||||||||||||||||||||||||||||||||||||||||||||||||||||||||||||||||||||||||||||||||||||||||||||||||||||||||||||||||||||||||||||||||||||||||||||||||||||||||||||||||||||||||||||||||||||||||||||||||||||||||||||||||||||||||||||||||||||||||
|
Point 47: |
Could the authors explain better the meaning of allocated? |
||||||||||||||||||||||||||||||||||||||||||||||||||||||||||||||||||||||||||||||||||||||||||||||||||||||||||||||||||||||||||||||||||||||||||||||||||||||||||||||||||||||||||||||||||||||||||||||||||||||||||||||||||||||||||||||||||||||||||||||||||||||||||||||||||||||||||||||
|
Response 47: |
Original contents: Line 394: The photosynthetic electron transport was more allocated to photorespiration with the decrease of the chlorophyll content Modified: The allocation of photosynthetic electron transport to photorespiration was enhanced with the decrease of the chlorophyll content. |
||||||||||||||||||||||||||||||||||||||||||||||||||||||||||||||||||||||||||||||||||||||||||||||||||||||||||||||||||||||||||||||||||||||||||||||||||||||||||||||||||||||||||||||||||||||||||||||||||||||||||||||||||||||||||||||||||||||||||||||||||||||||||||||||||||||||||||||
|
Point 48: |
The authors should underline the effect of H2S addition on PGP |
||||||||||||||||||||||||||||||||||||||||||||||||||||||||||||||||||||||||||||||||||||||||||||||||||||||||||||||||||||||||||||||||||||||||||||||||||||||||||||||||||||||||||||||||||||||||||||||||||||||||||||||||||||||||||||||||||||||||||||||||||||||||||||||||||||||||||||||
|
Response 48: |
Original contents: Line 396: In this study, NaCl treatment strongly up-regulated PGP, which indicated allocation of photosynthetic electron transport to Calvin cycle was reduced (Wingler et al. 2000). Modified: In this study, NaCl treatment strongly up-regulated PGP, which indicated allocation of photosynthetic electron transport to Calvin cycle was reduced (Wingler et al. 2000). The accumulation of PGP had no significantly different from control treatment. |
||||||||||||||||||||||||||||||||||||||||||||||||||||||||||||||||||||||||||||||||||||||||||||||||||||||||||||||||||||||||||||||||||||||||||||||||||||||||||||||||||||||||||||||||||||||||||||||||||||||||||||||||||||||||||||||||||||||||||||||||||||||||||||||||||||||||||||||
|
Point 49: |
It is significant the difference between the two ratios? As I wrote before, it is important to know the statistic relevance of the difference between NaCl and NaCl+H2S treatments |
||||||||||||||||||||||||||||||||||||||||||||||||||||||||||||||||||||||||||||||||||||||||||||||||||||||||||||||||||||||||||||||||||||||||||||||||||||||||||||||||||||||||||||||||||||||||||||||||||||||||||||||||||||||||||||||||||||||||||||||||||||||||||||||||||||||||||||||
|
Response 49: |
Original contents: Line 400: ATP synthase CF1 epsilon subunit (atpC, spot 7) is involved in energy supply needed for carbon dioxide reduction and fixation processes. Compared with NaCl treatment, atpC were recovered under NaCl+H2S treatment, implying the energy used for photosynthetic carbon assimilation was recovered with NaHS addition. Modified: ATP synthase CF1 epsilon subunit (atpC, spot 7) is involved in energy supply needed for carbon dioxide reduction and fixation processes. Although the accumulation of atpC was down-regulated, the accumulation of atpC was modified relative to NaCl treatment, implying the NaHS addition might have a positive effect to supply the energy. |
||||||||||||||||||||||||||||||||||||||||||||||||||||||||||||||||||||||||||||||||||||||||||||||||||||||||||||||||||||||||||||||||||||||||||||||||||||||||||||||||||||||||||||||||||||||||||||||||||||||||||||||||||||||||||||||||||||||||||||||||||||||||||||||||||||||||||||||
|
Point 50: |
Change as: triosephosphate isomerase (TPI, spots 16), and phosphoglycerate kinase (PGK, spot 35) were found to be down-regulated or unchanged under high salinity, respectively |
||||||||||||||||||||||||||||||||||||||||||||||||||||||||||||||||||||||||||||||||||||||||||||||||||||||||||||||||||||||||||||||||||||||||||||||||||||||||||||||||||||||||||||||||||||||||||||||||||||||||||||||||||||||||||||||||||||||||||||||||||||||||||||||||||||||||||||||
|
Response 50: |
Original contents: Line 405: triosephosphate isomerase (TPI, spots 16), and phosphoglycerate kinase (PGK, spot 35) were found to be unchanged or down-regulated under high salinity. Modified: triosephosphate isomerase (TPI, spots 16), and phosphoglycerate kinase (PGK, spot 35) were found to be down-regulated or unchanged under high salinity, respectively. |
||||||||||||||||||||||||||||||||||||||||||||||||||||||||||||||||||||||||||||||||||||||||||||||||||||||||||||||||||||||||||||||||||||||||||||||||||||||||||||||||||||||||||||||||||||||||||||||||||||||||||||||||||||||||||||||||||||||||||||||||||||||||||||||||||||||||||||||
|
Point 51: |
Change as: positively |
||||||||||||||||||||||||||||||||||||||||||||||||||||||||||||||||||||||||||||||||||||||||||||||||||||||||||||||||||||||||||||||||||||||||||||||||||||||||||||||||||||||||||||||||||||||||||||||||||||||||||||||||||||||||||||||||||||||||||||||||||||||||||||||||||||||||||||||
|
Response 51: |
Original contents: Line 418: Heat-shock protein (HSP, spot 19) and 20 kDa chaperonin family protein (Cpn 20, spot 26), which involved in protein synthesis, were found positivity modulated by NaHS addition. Modified: Heat-shock protein (HSP, spot 19) and 20 kDa chaperonin family protein (Cpn 20, spot 26) were involved in protein synthesis, which were positively modulated by NaHS addition. |
||||||||||||||||||||||||||||||||||||||||||||||||||||||||||||||||||||||||||||||||||||||||||||||||||||||||||||||||||||||||||||||||||||||||||||||||||||||||||||||||||||||||||||||||||||||||||||||||||||||||||||||||||||||||||||||||||||||||||||||||||||||||||||||||||||||||||||||
|
Point 52: |
I would suggest to link ATP synthase with oxidative stress too, since it has been widely discussed its involvement in this phenomenon [Houstek J, Pícková A, Vojtísková A, Mrácek T, Pecina P, Jesina P. (2006). Mitochondrial diseases and genetic defects of ATP synthase. Biochim Biophys Acta. 1757(9-10):1400-5]. Moreover, recently it has been predicted a protein network in which ATP synthase is involved, that can contribute to protect the plant against photo-oxidative damage [Bertini L, Palazzi L, Proietti S, Pollastri S, Arrigoni G, de Laureto P, Caruso C (2019). Proteomic Analysis of MeJa-Induced Defense Responses in Rice against Wounding. IJMS, 20, 2525]. |
||||||||||||||||||||||||||||||||||||||||||||||||||||||||||||||||||||||||||||||||||||||||||||||||||||||||||||||||||||||||||||||||||||||||||||||||||||||||||||||||||||||||||||||||||||||||||||||||||||||||||||||||||||||||||||||||||||||||||||||||||||||||||||||||||||||||||||||
|
Response 52: |
Original contents: Line 425: The ATP synthase is an important enzyme which catalyzes energy production by synthesizing ATP from ADP (Yasuda et al. 2001), providing energy to the TCA cycle. NaCl+H2S treatment significantly up-regulated the accumulation of ATP5PD and ATP5F1B. Modified: The ATP synthase is an important enzyme which catalyzes energy production by synthesizing ATP from ADP (Yasuda et al. 2001), providing energy to the TCA cycle. In mammals, the increased oxidative stress represents an important factor for isolated disorders of ATP synthase (Houštěk et al. 2006). It has connection between ATP synthase and oxidative stress. Moreover, recently it has been predicted a protein network in which ATP synthase is involved, that can contribute to protect the plant against photo-oxidative damage (Bertini et al. 2019). NaCl+H2S treatment significantly up-regulated the accumulation of ATP5PD and ATP5F1B. |
||||||||||||||||||||||||||||||||||||||||||||||||||||||||||||||||||||||||||||||||||||||||||||||||||||||||||||||||||||||||||||||||||||||||||||||||||||||||||||||||||||||||||||||||||||||||||||||||||||||||||||||||||||||||||||||||||||||||||||||||||||||||||||||||||||||||||||||
|
Page 15 |
|||||||||||||||||||||||||||||||||||||||||||||||||||||||||||||||||||||||||||||||||||||||||||||||||||||||||||||||||||||||||||||||||||||||||||||||||||||||||||||||||||||||||||||||||||||||||||||||||||||||||||||||||||||||||||||||||||||||||||||||||||||||||||||||||||||||||||||||
|
Point 53: |
For H2O2 and MDA the decrease is not statistically significant. Please change the text acciordingly to this evidence |
||||||||||||||||||||||||||||||||||||||||||||||||||||||||||||||||||||||||||||||||||||||||||||||||||||||||||||||||||||||||||||||||||||||||||||||||||||||||||||||||||||||||||||||||||||||||||||||||||||||||||||||||||||||||||||||||||||||||||||||||||||||||||||||||||||||||||||||
|
Response 53: |
Original contents: Line 439: In the present study, NaHS pretreatment reduced the accumulation of H2O2, MDA, and REL under salt stress, which is in agreement with the previous study by Shi et al. (2013) Modified: In the present study, NaHS pretreatment reduced the accumulation of REL under salt stress, which is in agreement with the previous study by Shi et al. (2013). |
||||||||||||||||||||||||||||||||||||||||||||||||||||||||||||||||||||||||||||||||||||||||||||||||||||||||||||||||||||||||||||||||||||||||||||||||||||||||||||||||||||||||||||||||||||||||||||||||||||||||||||||||||||||||||||||||||||||||||||||||||||||||||||||||||||||||||||||
|
Point 54: |
Results are not significant for APX and SOD. Please change the text acciordingly to this evidence |
||||||||||||||||||||||||||||||||||||||||||||||||||||||||||||||||||||||||||||||||||||||||||||||||||||||||||||||||||||||||||||||||||||||||||||||||||||||||||||||||||||||||||||||||||||||||||||||||||||||||||||||||||||||||||||||||||||||||||||||||||||||||||||||||||||||||||||||
|
Response 54: |
Original contents: Line 446: In this study, compared with NaCl treatment, the activity of superoxide dismutase (SOD) and cytosolic ascorbate peroxidase (APX), and the content of GSH were found to be induced by application of H2S (Figure 2). Modified: In this study, compared with NaCl treatment, the content of GSH were found to be induced by application of H2S (Figure 2). |
||||||||||||||||||||||||||||||||||||||||||||||||||||||||||||||||||||||||||||||||||||||||||||||||||||||||||||||||||||||||||||||||||||||||||||||||||||||||||||||||||||||||||||||||||||||||||||||||||||||||||||||||||||||||||||||||||||||||||||||||||||||||||||||||||||||||||||||
|
Point 55: |
Actually, the AsA-GSH cycle produces GSSG, not GSH. Thus, this hypothesis should be revised. |
||||||||||||||||||||||||||||||||||||||||||||||||||||||||||||||||||||||||||||||||||||||||||||||||||||||||||||||||||||||||||||||||||||||||||||||||||||||||||||||||||||||||||||||||||||||||||||||||||||||||||||||||||||||||||||||||||||||||||||||||||||||||||||||||||||||||||||||
|
Response 55: |
Original contents: Line 448: As an electron donor, APX reduces H2O2 into the water using AsA, and monodehydroascorbate can react with GSH to produce AsA and oxidized GSH catalyzed by DHAR (Suo et al. 2015). Modified: As an electron donor, APX reduces H2O2 into the water using AsA, and monodehydroascorbate can react with GSH to produce AsA and GSSG catalyzed by DHAR (Suo et al. 2015). |
||||||||||||||||||||||||||||||||||||||||||||||||||||||||||||||||||||||||||||||||||||||||||||||||||||||||||||||||||||||||||||||||||||||||||||||||||||||||||||||||||||||||||||||||||||||||||||||||||||||||||||||||||||||||||||||||||||||||||||||||||||||||||||||||||||||||||||||
|
Point 56: |
Could the authors explain the difference between the two SOD enzymes? |
||||||||||||||||||||||||||||||||||||||||||||||||||||||||||||||||||||||||||||||||||||||||||||||||||||||||||||||||||||||||||||||||||||||||||||||||||||||||||||||||||||||||||||||||||||||||||||||||||||||||||||||||||||||||||||||||||||||||||||||||||||||||||||||||||||||||||||||
|
Response 56: |
Original contents: Line 453: Moreover, copper/zinc superoxide dismutase CSD2A-1 (CSD2, spot 17) was found up-regulated under NaCl+H2S treatment, but SOD (spot 18) showed the opposite trend. We propose that CSD2 …… Modified: Moreover, copper/zinc superoxide dismutase CSD2A-1 (CSD2, spot 17) was found up-regulated under NaCl+H2S treatment, but SOD (spot 18) showed the opposite trend. The SODs were divided into four types according to the different metal cofactors in the catalytic site and CSD2 is mainly found in higher plants (Abreu and Cabelli 2010). We propose that CSD2 …… |
||||||||||||||||||||||||||||||||||||||||||||||||||||||||||||||||||||||||||||||||||||||||||||||||||||||||||||||||||||||||||||||||||||||||||||||||||||||||||||||||||||||||||||||||||||||||||||||||||||||||||||||||||||||||||||||||||||||||||||||||||||||||||||||||||||||||||||||
|
Point 57: |
Please, revise the conclusion on the base of the comments inside the text |
||||||||||||||||||||||||||||||||||||||||||||||||||||||||||||||||||||||||||||||||||||||||||||||||||||||||||||||||||||||||||||||||||||||||||||||||||||||||||||||||||||||||||||||||||||||||||||||||||||||||||||||||||||||||||||||||||||||||||||||||||||||||||||||||||||||||||||||
|
Response 57: |
Original contents: Line 456: In short, based on the above results, it can infer that H2S has a positive effect to protect the plant from the excess ROS. Modified: In short, based on the above results, it can infer that H2S increased non-enzymatic antioxidants to detoxify ROS and prevent cellular damage under high salinity. |
||||||||||||||||||||||||||||||||||||||||||||||||||||||||||||||||||||||||||||||||||||||||||||||||||||||||||||||||||||||||||||||||||||||||||||||||||||||||||||||||||||||||||||||||||||||||||||||||||||||||||||||||||||||||||||||||||||||||||||||||||||||||||||||||||||||||||||||
|
Point 58: |
Cysteine synthase is a pyridoxal phosphate-binding protein. Are the authors sure about the correspondence of pyridoxine biosynthesis PDX1-like protein 3 and cysteine synthase? The references seem to be not adequate. In the last paragraph, it is not clear the relation between CP and PDX1. The authors should explain better how their down/up-regulation could improve plant tolerance to NaCl. |
||||||||||||||||||||||||||||||||||||||||||||||||||||||||||||||||||||||||||||||||||||||||||||||||||||||||||||||||||||||||||||||||||||||||||||||||||||||||||||||||||||||||||||||||||||||||||||||||||||||||||||||||||||||||||||||||||||||||||||||||||||||||||||||||||||||||||||||
|
Response 58: |
Original contents: Line 466: PDX1 was identified as a cysteine synthase (Komatsu et al. 2014), inducing cysteine biosynthesis as a protective measure against high ion concentrations, or synthesis of cystatin to inhibit PCD (Youssefian et al. 1993). We observed that PDX1 was significantly up-regulated under NaCl+H2S treatment. Modified: PDX1 was regarded as pyridoxal phosphate-binding protein (Komatsu et al. 2014). Pyridoxal phosphate-binding proteins protected plants from high ion concentrations by inducing cysteine biosynthesis, or synthesis of cystatin to inhibit PCD (Youssefian et al. 1993). We observed that PDX1 was significantly up-regulated to inhibit PCD under NaCl+H2S treatment. |
||||||||||||||||||||||||||||||||||||||||||||||||||||||||||||||||||||||||||||||||||||||||||||||||||||||||||||||||||||||||||||||||||||||||||||||||||||||||||||||||||||||||||||||||||||||||||||||||||||||||||||||||||||||||||||||||||||||||||||||||||||||||||||||||||||||||||||||
|
Point 59: |
Moreover, which is the role of ADH in response to high salinity and NaCl+H2S treatments? |
||||||||||||||||||||||||||||||||||||||||||||||||||||||||||||||||||||||||||||||||||||||||||||||||||||||||||||||||||||||||||||||||||||||||||||||||||||||||||||||||||||||||||||||||||||||||||||||||||||||||||||||||||||||||||||||||||||||||||||||||||||||||||||||||||||||||||||||
|
Response 59: |
Line 466: ADH catalyzes the regeneration of NAD+ from the reduction of acetaldehyde to ethanol (Huang et al. 2002). H2S has been shown to inhibit ADH activity in roots of freshwater marsh species Spartina alterniflora (Koch et al. 1990), but the accumulation of ADH was down-regulated under NaCl treatment and had not changed significantly with the NaHS addition. We observed that H2S helped alleviate the reduction of ADH under NaCl treatment |
||||||||||||||||||||||||||||||||||||||||||||||||||||||||||||||||||||||||||||||||||||||||||||||||||||||||||||||||||||||||||||||||||||||||||||||||||||||||||||||||||||||||||||||||||||||||||||||||||||||||||||||||||||||||||||||||||||||||||||||||||||||||||||||||||||||||||||||
|
Point 60: |
bHLH is a protein motif. What do the authors mean? bHLH transcription factors? |
||||||||||||||||||||||||||||||||||||||||||||||||||||||||||||||||||||||||||||||||||||||||||||||||||||||||||||||||||||||||||||||||||||||||||||||||||||||||||||||||||||||||||||||||||||||||||||||||||||||||||||||||||||||||||||||||||||||||||||||||||||||||||||||||||||||||||||||
|
Response 60: |
Original contents: Line 470: The bHLH served as negative feedback regulatory loop in ABA signaling in Arabidopsis thaliana (Zheng et al. 2019). Modified: The bHLH transcription factors served as negative feedback regulatory loop in ABA signaling in Arabidopsis thaliana (Zheng et al. 2019). |
||||||||||||||||||||||||||||||||||||||||||||||||||||||||||||||||||||||||||||||||||||||||||||||||||||||||||||||||||||||||||||||||||||||||||||||||||||||||||||||||||||||||||||||||||||||||||||||||||||||||||||||||||||||||||||||||||||||||||||||||||||||||||||||||||||||||||||||
|
Point 61: |
With respect to NaCl treatment the expresssion level of Asr and bHLH145 upon NaCl+H2S treatment is lower. What do the authors mean with "recovered under NaCl+H2S treatment"? The sentence is completely ungrammatical. Please, reformulate it. |
||||||||||||||||||||||||||||||||||||||||||||||||||||||||||||||||||||||||||||||||||||||||||||||||||||||||||||||||||||||||||||||||||||||||||||||||||||||||||||||||||||||||||||||||||||||||||||||||||||||||||||||||||||||||||||||||||||||||||||||||||||||||||||||||||||||||||||||
|
Response 61: |
Original contents: Line 472: In the present study, the up-regulated of Asr and bHLH145 under salt treatment and recovered under NaCl+H2S treatment, which indicated the signal transduction of ABA was inhibited by the addition of NaHS. Modified: In the present study, Asr and bHLH145 were up-regulated under salt treatment and had not changed significantly under NaCl+H2S treatment, which indicated the signal transduction of ABA was inhibited by the addition of NaHS. |
||||||||||||||||||||||||||||||||||||||||||||||||||||||||||||||||||||||||||||||||||||||||||||||||||||||||||||||||||||||||||||||||||||||||||||||||||||||||||||||||||||||||||||||||||||||||||||||||||||||||||||||||||||||||||||||||||||||||||||||||||||||||||||||||||||||||||||||
|
Point 62: |
The physiological effect of the presence of ABA-responsive elements in the promoter regions of the genes encoding CP, should be explained. |
||||||||||||||||||||||||||||||||||||||||||||||||||||||||||||||||||||||||||||||||||||||||||||||||||||||||||||||||||||||||||||||||||||||||||||||||||||||||||||||||||||||||||||||||||||||||||||||||||||||||||||||||||||||||||||||||||||||||||||||||||||||||||||||||||||||||||||||
|
Response 62: |
Original contents: Line 475: Furthermore, in the promoter regions of the genes encoding CP exists ABA-responsive elements (Szewińska et al. 2016). Modified: Furthermore, in the promoter regions of the genes encoding CP exists ABA-responsive elements, transcription of CP is repressed by ABA through DOF (DNA binding One Zinc Finger) (Szewińska et al. 2016). |
||||||||||||||||||||||||||||||||||||||||||||||||||||||||||||||||||||||||||||||||||||||||||||||||||||||||||||||||||||||||||||||||||||||||||||||||||||||||||||||||||||||||||||||||||||||||||||||||||||||||||||||||||||||||||||||||||||||||||||||||||||||||||||||||||||||||||||||
|
Point 63: |
The conclusions should be revised on the base of the comments in the text |
||||||||||||||||||||||||||||||||||||||||||||||||||||||||||||||||||||||||||||||||||||||||||||||||||||||||||||||||||||||||||||||||||||||||||||||||||||||||||||||||||||||||||||||||||||||||||||||||||||||||||||||||||||||||||||||||||||||||||||||||||||||||||||||||||||||||||||||
|
Response 63: |
Original contents: (3) H2S increased enzymatic and non-enzymatic antioxidants to detoxify ROS and prevent cellular damage under high salinity; Modified: (3) H2S increased non-enzymatic antioxidants to detoxify ROS and prevent cellular damage under high salinity; |
||||||||||||||||||||||||||||||||||||||||||||||||||||||||||||||||||||||||||||||||||||||||||||||||||||||||||||||||||||||||||||||||||||||||||||||||||||||||||||||||||||||||||||||||||||||||||||||||||||||||||||||||||||||||||||||||||||||||||||||||||||||||||||||||||||||||||||||
|
Point 64: |
Chanhe as: affected |
||||||||||||||||||||||||||||||||||||||||||||||||||||||||||||||||||||||||||||||||||||||||||||||||||||||||||||||||||||||||||||||||||||||||||||||||||||||||||||||||||||||||||||||||||||||||||||||||||||||||||||||||||||||||||||||||||||||||||||||||||||||||||||||||||||||||||||||
|
Response 64: |
Original contents: (4) H2S effected the ABA signaling pathway, which led to sustained plant adaptation to high salinity. Modified: (4) H2S affected the ABA signaling pathway, which led to sustained plant adaptation to high salinity. |
||||||||||||||||||||||||||||||||||||||||||||||||||||||||||||||||||||||||||||||||||||||||||||||||||||||||||||||||||||||||||||||||||||||||||||||||||||||||||||||||||||||||||||||||||||||||||||||||||||||||||||||||||||||||||||||||||||||||||||||||||||||||||||||||||||||||||||||
|
Point 65: |
The figure should be revised on the base of the comments in the text |
||||||||||||||||||||||||||||||||||||||||||||||||||||||||||||||||||||||||||||||||||||||||||||||||||||||||||||||||||||||||||||||||||||||||||||||||||||||||||||||||||||||||||||||||||||||||||||||||||||||||||||||||||||||||||||||||||||||||||||||||||||||||||||||||||||||||||||||
|
Response 65: |
We modified the figure. | ||||||||||||||||||||||||||||||||||||||||||||||||||||||||||||||||||||||||||||||||||||||||||||||||||||||||||||||||||||||||||||||||||||||||||||||||||||||||||||||||||||||||||||||||||||||||||||||||||||||||||||||||||||||||||||||||||||||||||||||||||||||||||||||||||||||||||||||
|
Point 66: |
What do the double arrow stand for? |
||||||||||||||||||||||||||||||||||||||||||||||||||||||||||||||||||||||||||||||||||||||||||||||||||||||||||||||||||||||||||||||||||||||||||||||||||||||||||||||||||||||||||||||||||||||||||||||||||||||||||||||||||||||||||||||||||||||||||||||||||||||||||||||||||||||||||||||
|
Response 66: |
The left and right arrow stand for NaCl and NaCl+H2S treatment, respectively. The red color arrow indicates physiological indexes increased change and the green color arrow indicates physiological indexes decreased change.
|
||||||||||||||||||||||||||||||||||||||||||||||||||||||||||||||||||||||||||||||||||||||||||||||||||||||||||||||||||||||||||||||||||||||||||||||||||||||||||||||||||||||||||||||||||||||||||||||||||||||||||||||||||||||||||||||||||||||||||||||||||||||||||||||||||||||||||||||
